# PHA-4/FoxA senses nucleolar stress to regulate lipid accumulation in *Caenorhabditis elegans*

Jieyu Wu[1,2], Xue Jiang[1,2], Yamei Li[1,3], Tingting Zhu[1,2], Jingjing Zhang[1,4], Zhiguo Zhang[1,2], Linqiang Zhang[1], Yuru Zhang[1], Yanli Wang[1], Xiaoju Zou[5] & Bin Liang[1,4]

The primary function of the nucleolus is ribosome biogenesis, which is an extremely energetically expensive process. Failures in ribosome biogenesis cause nucleolar stress with an altered energy status. However, little is known about the underlying mechanism linking nucleolar stress to energy metabolism. Here we show that nucleolar stress is triggered by inactivation of RSKS-1 (ribosomal protein S6 kinase), RRP-8 (ribosomal RNA processing 8), and PRO-2/3 (proximal proliferation), all of which are involved in ribosomal RNA processing or inhibition of rDNA transcription by actinomycin D (AD), leading to excessive lipid accumulation in *Caenorhabditis elegans*. The transcription factor PHA-4/FoxA acts as a sensor of nucleolar stress to bind to and transactivate the expression of the lipogenic genes *pod-2* (acetyl-CoA carboxylase), *fasn-1* (fatty acid synthase), and *dgat-2* (diacylglycerol *O*-acyltransferase 2), consequently promoting lipid accumulation. Importantly, inactivation of *pha-4* or *dgat-2* is sufficient to abolish nucleolar stress-induced lipid accumulation and prolonged starvation survival. The results revealed a distinct PHA-4-mediated lipogenesis pathway that senses nucleolar stress and shifts excessive energy for storage as fat.

[1] Key Laboratory of Animal Models and Human Disease Mechanisms of the Chinese Academy of Sciences and Yunnan Province, Center for Excellence in Animal Evolution and Genetics, Kunming Institute of Zoology, Chinese Academy of Sciences, Kunming 650223, China. [2] Kunming College of Life Science, University of Chinese Academy of Sciences, Kunming 650204, China. [3] School of Life Science, University of Science and Technology of China, Hefei 230027, China. [4] Department of Hepatobiliary Surgery, Affiliated Hospital of Guangdong Medical University, Zhanjiang 524001, China. [5] Key Laboratory of Special Biological Resource Development and Utilization of University in Yunnan Province, Department of Life Science and Biotechnology, Kunming University, Kunming 650214, China. These authors contributed equally: Jieyu Wu, Xue Jiang. Correspondence and requests for materials should be addressed to X.Z. (email: xiaojuzou@163.com) or to B.L. (email: liangb@mail.kiz.ac.cn)

In eukaryotic cells, the primary function of the nucleolus is the rapid production of small and large ribosome subunits, a process that generally includes ribosomal RNA (rRNA) transcription and processing, as well as ribosome subunit assembly[1]. In the mammalian nucleolus, rDNA genes are transcribed by RNA polymerase I into 47S pre-rRNA, which is subsequently processed and modified by small nucleolar ribonucleoproteins to generate 28S, 18S, and 5.8S rRNAs. These rRNAs are subsequently assembled with ribosomal proteins (RPs) to form small and large preribosome subunits, which are exported to the cytoplasm to eventually form the mature 40S and 60S ribosome subunits[2,3].

Failures in ribosome biogenesis or function result in a condition termed nucleolar stress, which ultimately leads to disruptions in cell homeostasis[4,5]. Extensive studies have reported a well-known p53-dependent pathway that regulates cell cycle arrest or apoptosis in response to nucleolar stress[5]. Nucleolar stress stimulates RPs (RpL5 and RpL11) to bind to MDM2 (murine and/or human double minute 2), which encodes the ubiquitin E3 ligase that negatively regulates p53, and thereby disrupts its association with P53, leading to p53 induction and cell cycle arrest or apoptosis[6,7]. However, more than half of human cancers lack functional p53, non-mammalian systems such as yeast lack p53, and both *Caenorhabditis elegans* and *Drosophila* lack MDM2, although these organisms possess p53. These data argue that other p53-independent nuclear stress pathways may be evolutionarily conserved or markedly vary across eukaryotes.

Nearly all metabolic and signaling pathways ultimately lead to or from the nucleolus. Ribosome biogenesis is the most energy-consuming process within proliferating eukaryotic cells. This process must be tightly regulated and rapidly adaptable to respond to metabolic and environmental changes. An increased NAD +/NADH ratio promotes the repression of rRNA synthesis and promotes the restoration of energy balance by the protein complex energy-dependent nucleolar silencing complex (eNoSC)[8]. Activation of hepatic rRNA transcription in mice promotes hepatic energy consumption to reduce lipid accumulation in the liver[9]. The RP-MDM2-P53 pathway is also critical for sensing nutrient deprivation and maintaining liver lipid homeostasis. The Mdm2C305F mutation promotes lipid accumulation under normal feeding conditions and hepatosteatosis under acute fasting conditions[10]. Both $p53^{-/-}$ MEF cells and the livers of $p53^{-/-}$ mice accumulate more lipid droplets (LDs) than their wild-type (WT) controls under normal conditions[10]. In addition, eNoSC may act as a sensor in the nucleolus connecting intracellular energy status with p53 activation[11,12]. Collectively, these studies demonstrate that nucleolar stress triggered by the perturbation of ribosome biogenesis can force the cell to shift its energy status and eventually alter lipid homeostasis. However, the underlying mechanisms linking nucleolar stress and lipid accumulation remain largely unknown.

Many aspects of nucleolar function and ribosome biogenesis are conserved within eukaryotic organisms, from yeast to humans. The model organism *C. elegans* contains many genes that are involved in ribosome biogenesis, such as *rsks-1* encoding a homolog of S6K that positively regulates several steps in ribosome biogenesis[13], as well as the *pro-1*, *pro-2*, and *pro-3* genes, which have been implicated in rRNA processing and ribosome assembly[14]. In the present study, we first identify a mutation in *rrp-8*, which encodes an ortholog of rRNA processing 8 (RRP8) and leads to an impaired ribosome profile and elevated lipid accumulation. Furthermore, we showed that nucleolar stress induced by genetic mutations in *rsks-1*, *pro-2*, and *pro-3*, as well as inhibition of rDNA transcription using actinomycin D (AD), leads to excessive lipid accumulation. Importantly, we observed that the transcription factor PHA-4/FoxA acts as a sensor of

nucleolar stress to transactivate the expression of the lipogenic genes *pod-2*, *fasn-1*, and *dgat-2*, leading to lipid accumulation and extended survival under starvation conditions.

## Results

**Mutation of *rrp-8* leads to excessive lipid accumulation.** In a forward genetic screen of fat regulators using the mutagen ethyl methane sulfonate (EMS) in *C. elegans*, we isolated the mutant *kun54*, which shows altered lipid accumulation. The Nile Red, LipidTox Red, and Oil Red O dyes have been shown to be able to stain neutral lipids of post-fixed worms[15,16]. In comparison with WT worms, worms containing the *kun54* mutation displayed enlarged LDs and increased lipid accumulation, as indicated by the post fixation of Nile Red staining (Fig. 1a), LipidTox Red staining (Fig. 1b), and Oil Red O staining (Fig. 1c). Quantification of the LD size from Nile Red staining of fixed worms showed that *kun54* mutant worms had an increased percentage of larger LDs (> 3 μm) but a decreased percentage of smaller lipid droplets (< 1 μm) than those of WT worms (Fig. 1d). In addition, lipid analysis by thin layer chromatography and gas chromatography (TLC/GC) consistently showed that the triacylglycerol (TAG) content was apparently increased from 49% in WT to 58% in the *kun54* mutant (Fig. 1e).

To determine the specific gene mutation in the *kun54* mutant, we performed rapid single-nucleotide polymorphism (SNP) mapping[17] combined with whole-genome sequencing[18]. The *kun54* mutation was initially mapped to chromosome IV following the methodology by Davis et al.[17] (Supplementary Fig. 1A), and subsequently mapped to 0.4 M region near SNP Y66H1A (−26) at the end of chromosome IV by interval mapping (Fig. 1f and Supplementary Fig. 1B–D). Whole-genome sequencing analysis of the *kun54* mutant further revealed a G to A substitution in the fifth exon of T07A9.8 in this region on chromosome IV, replacing the glycine from WT with an arginine in the *kun54* mutant (G301R) (Fig. 1f). T07A9.8 is predicted to encode an ortholog of RRP8 (www.wormbase.org) and we thereby named T07A9.8 as RRP-8 in *C. elegans*. To further confirm the above phenotypes of the *rrp-8(kun54)* mutant, we used CRISPR/cas-9 technology to generate a 42 bp deletion (*kun122*) in the second exon of the *rrp-8* gene (Fig. 1g). Similar to the *rrp-8(kun54)* mutant, the *rrp-8(kun122)* mutant also showed an increased lipid accumulation (Fig. 1g) and LD size (Fig. 1h). Therefore, these results consistently demonstrated that the disruption of *rrp-8* leads to excessive lipid accumulation.

To explore RRP-8 expression, we generated a transgenic strain for the translational expression of *rrp-8::gfp* {WT;kunEx121[Prrp-8::rrp-8::GFP]}. The yeast homolog RRP8 has been reported to be a nucleolar protein[19,20]. The green fluorescence of RRP-8::GFP was clearly observed in the nuclei of cells by initial DAPI (4′,6-diamidino-2-phenylindole) nuclear staining (Fig. 1i, upper panel) and completely colocalized with mCherry::FIB-1 (Fig. 1i, lower panel) encoding the nucleolar protein fibrillarin, suggesting that RRP-8 was expressed in the nucleolus in *C. elegans*. Moreover, RRP-8::GFP was ubiquitously expressed in almost all cells throughout all developmental stages from embryo to adulthood (Fig. 1j). Interestingly, the G301R mutation of RRP-8 showed an obvious decrease in expression (Supplementary Fig. 2A), which was indistinguishable from the nucleolar localization compared with RRP-8::GFP (Supplementary Fig. 2B).

**Inactivation of RRP-8 impairs pre-rRNA processing.** Previous studies in yeast have reported that nucleolar protein RRP8 displays methyltransferase activity for the m1A base modification of 25S rRNA and has been implicated in pre-rRNA cleavage at site A2[19,20] (Fig. 2a). Its mammalian homolog nucleomethylin has

also been reported to modify rRNA[20,21], depending on its methyltransferase-like domain (MLD)[8]. Similarly, sequence analysis revealed that *C. elegans* RRP-8 also contained an MLD near the carboxyl terminus (Fig. 2b). Interestingly, the mutated glycine residue (Gly, G) in MLD was highly conserved in RRP8 proteins from *C. elegans* to mammals (Fig. 2b). To examine whether *C. elegans* RRP-8 might have functions similar to its yeast and mammalian homologs, we measured the methylation rate at A674 in 26S rRNA, a conserved site corresponding to A645 in 25S rRNA in yeast (Fig. 2c). Indeed, the methylation rate at A674 in

26S rRNA was decreased in the *rrp-8(kun54)* mutant in comparison with WT (Fig. 2d), suggesting that this conserved glycine residue (G301) is crucial for RRP-8 as a methyltransferase.

Next, to examine whether the G301R mutation of RRP-8 also affected pre-rRNA cleavage in *C. elegans* (Fig. 2e), we measured the pre-rRNA levels in the *rrp-8(kun54)* mutant by quantitative PCR (QPCR). The pre-rRNA levels showed no difference between the *rrp-8(kun54)* mutant and WT (Fig. 2f), whereas the rRNA level of site III that corresponds to site A2 in yeast among nine rRNA probes displayed a slight increase in the *rrp-8(kun54)*

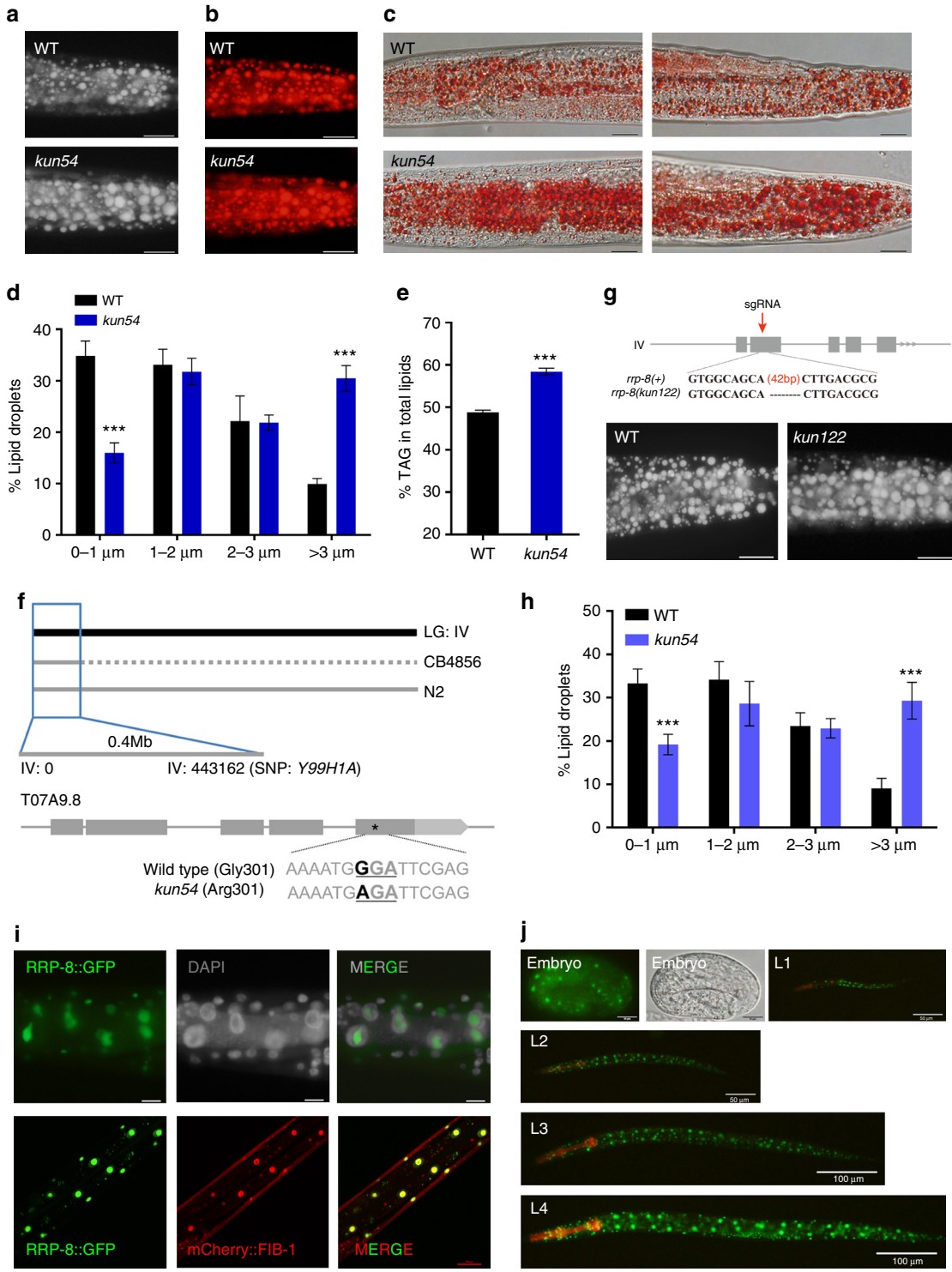

mutant compared with WT (Fig. 2g), indicating a reduction of the cleavage efficiency of pre-RNA when RRP-8 was disrupted.

To further confirm the above results, we performed northern blotting to analyze rRNA processing. The processed rRNA intermediates were analyzed based on previous reports[14,22]. To detect the levels of pre-RNA (a) and rRNA intermediates (b, b1, c, and c1) in WT and rrp-8(kun54) mutant worms, we designed three probes (1–3) corresponding to distinct regions of the pre-RNA (Fig. 2e) following the report by Voutev et al.[23]. Equal amounts of total RNA isolated from WT and rrp-8(kun54) were loaded on a denaturing gel, which showed that the levels of 26S and 18S were not affected (Fig. 2h, left panel). Northern hybridization with probes 1–3 all indicated that pre-rRNA (band a) presented invariable levels in both WT and rrp-8(kun54) (Fig. 2h). In addition, hybridization with probe 1 indicated that the total levels of rRNA processing intermediates b + b1 were unchanged (Fig. 2h). Importantly, the level of band b containing site III, which was specifically recognized by probe 2 (Fig. 2e), was also elevated in rrp-8(kun54) compared with WT worms (Fig. 2h), suggesting that the cleavage efficiency of site III that corresponds to site A2 in yeast was affected by the dysfunction of RRP-8. Altogether, these data consistently support that RRP-8 functions as an evolutionarily conserved methyltransferase that is involved in rRNA processing in yeast, C. elegans, and mammals.

**Nucleolar stress elicits excessive lipid accumulation**. The primary function of the eukaryotic nucleolus is ribosome biosynthesis. Failures in ribosome biogenesis or function cause nucleolar stress, ultimately leading to disruptions in cell homeostasis[4,5]. As the nucleolar protein RRP-8 participates in rRNA processing and the inactivation of RRP-8 leads to excessive lipid accumulation, we questioned whether nucleolar stress triggered by perturbation of ribosome biogenesis via gene mutations or chemical compounds might yield common effects on lipid accumulation.

The process of ribosomal biogenesis in the nucleolus is highly conserved, well characterized, and generally includes rDNA transcription and pre-rRNA processing, as well as assembly in eukaryotic organisms (Fig. 3a)[24]. Ribosomal profile analysis showed that the rrp-8(kun54) mutant displayed decreased polysome levels (Fig. 3b and c), consistently supporting a role for RRP-8 in rRNA processing (Fig. 2d and g). AD is an antibiotic compound that preferentially inhibits rDNA transcription[25,26]. Consistently, the above results showing the altered ribosome profile were also observed in AD-treated worms (Fig. 3b and c). Remarkably, similar to rrp-8(kun54) mutant worms, AD-treated worms showed elevated lipid accumulation, as indicated by Nile Red staining of fixed worms (Fig. 3d) and an increase in larger LDs (> 3 μm) (Fig. 3e) and TAG content based on TLC/GC

analysis (57.29 ± 0.45 vs 49.31 ± 0.39 in WT, % TAG in total lipids) (Fig. 3a). AD treatment did not further enhance lipid accumulation in rrp-8(kun54) mutants (Fig. 3d and e), suggesting that RRP-8 and AD might act in one pathway.

In C. elegans, the pro-1, pro-2, and pro-3 genes have been implicated in rRNA processing and ribosome assembly[14]. Similarly, two available mutants, pro-2(na27) and pro-3(ar226), also showed increased fat contents (58.45 ± 0.51 and 57.36 ± 0.39, respectively) compared with WT worms according to the TLC/GC analysis (Fig. 3a). The mammalian target of rapamycin complex 1 (mTORC1) regulates ribosome biogenesis[27]. In C. elegans, rsks-1 is a homolog of the p70 RP S6 kinase, an effector of the TOR pathway. Previous studies have shown that S6K controls many nucleolar proteins associated with ribosomal biogenesis[28]. Consistent with the above results (Fig. 3a), the rsks-1(ok1255) mutant displayed not only decreased polysome levels (Fig. 3b) but also increased lipid accumulation (Fig. 3d–h), whereas the rsks-1 (ok1255) mutation did not further affect lipid accumulation in the rrp-8(kun54) mutant or in AD-treated worms (Fig. 3d–h). Altogether, these findings indicate that nucleolar stress triggered by the perturbation of ribosomal biogenesis results in excessive lipid accumulation.

**PHA-4 senses nucleolar stress to promote lipid accumulation**. As nucleolar stress leads to excessive lipid accumulation, we explored the intracellular factor responsible for nucleolar stress. The tumor suppressor P53 has been extensively shown to have an essential role in the response to nucleolar stress[4,5]. cep-1 is an ortholog of mammalian p53 in C. elegans. We observed that the fluorescence intensity of CEP-1::GFP {gtIs1[cep-1::gfp]} was indeed elevated not only in rrp-8(kun54) and rsks-1(ok1255) mutants but also in AD-treated worms (Supplementary Fig. 3A and B). However, neither of the two mutant alleles (gk138 and lg12501) of cep-1 could suppress the increased lipid accumulation in the rrp-8(kun54) and rsks-1(ok1255) mutants, as indicated by Nile Red staining of fixed worms (Supplementary Fig. 3C) and quantification by TCL/GC analysis (Supplementary Fig. 3D). Therefore, these results indicate that P53/CEP-1 is not required for nucleolar stress-induced lipid accumulation, although its expression increases in response to nucleolar stress.

The mTOR/S6K pathway can integrate many nutrients and growth factors to control mammalian rRNA synthesis[29]. mTORC1 can activate the transcription factor sterol regulatory element-binding protein 1 (SREBP1), a master regulator of lipogenesis, to promote lipid synthesis[30]. However, the fluorescence intensity of GFP::SBP-1 {ftIs7[Psbp-1::gfp::sbp-1]}, a homolog of SREBPs[31,32], was indistinguishable between the rrp-8 (kun54) mutant and WT worms (Supplementary Fig. 4A). In addition, the insulin/insulin growth factor 1 (IGF-1) signaling

**Fig. 1** Mutation of kun54 leads to excessive lipid accumulation. **a–c** rrp-8(kun54) is a mutant strain obtained from a forward genetic screen of fat regulators by EMS mutagenesis. Compared with wild-type (WT) worms, rrp-8(kun54) worms consistently show increased lipid accumulation by **a** Nile Red staining, **b** LipidTox Red staining, and **c** Oil Red staining of fixed worms. The stained particles are lipid droplets (LDs) in representative 1-day-old adult worms. Scale bar represents 20 μm. Magnification, × 400. **d** Distribution of the lipid droplet size (% lipid droplets), as measured from Nile Red staining of fixed worms from **a**. Data are presented as mean ± SD of 10 animals for each worm strain. **e** Lipid contents were quantified by TLC/GC and presented as the percentage of triacylglycerol (% TAG) in total lipids (TAG + phospholipids, PL). Data are presented as the mean ± SD of four biological repeats. **f** Genetic mapping of the kun54 mutation (the mapping details are shown in Supplementary Fig. 1) and the mutated position of kun54 in T07A9.8 is indicated by an asterisk. **g** Information for generation of the rrp-8(kun122) mutation. Top panel: schematic diagram of the generation of the kun122 mutation (42 bp deletion) using CRISPR/cas-9 technology. The red arrow indicates the site of the designed sgRNA. Bottom panel: Nile Red staining of fixed worms. **h** Distribution of the lipid droplet size (% lipid droplets), as measured in Nile Red-stained fixed worms from **g**. Data are presented as the mean ± SD of 10 animals for each worm strain. **i** Nuclear localization of RRP-8::GFP with DAPI (upper panel, fluorescence microscopy) and the nucleolar marker mCherry::FIB-1 (lower panel, confocal microscopy). Scale bar represents 10 μm in upper panel and 20 μm in lower panel. **j** Fluorescence microscopy of nuclear expression of RRP-8::GFP {kunEx121 [Prrp-8::rrp-8::gfp]} throughout different developmental stages (embryo to L4 larvae). In all of the represented animals, the anterior is indicated on the left and posterior is indicated on the right. Significant difference between WT and a specific worm strain, Student's t-test, ***P < 0.001

pathway has an important role in regulating stress responses, metabolism, and the abundance of ribosomal subunits in *C. elegans*[33,34]. In contrast, the expression of DAF-16::GFP {*zIs345 [daf-16::gfp]*}, a homolog of the FOXO transcription factor in insulin/IGF-1 signaling pathway[35], showed no differences between the *rrp-8(kun54)* mutant and WT worms

(Supplementary Fig. 4B). Both the *sbp-1(ep79)* and *daf-16 (mu86)* mutants were incapable of suppressing the excessive lipid accumulation in the *rrp-8(kun54)* mutant (Supplementary Fig. 4C and D). Altogether, these results suggest that nucleolar stress-induced lipid accumulation may not be dependent on these factors.

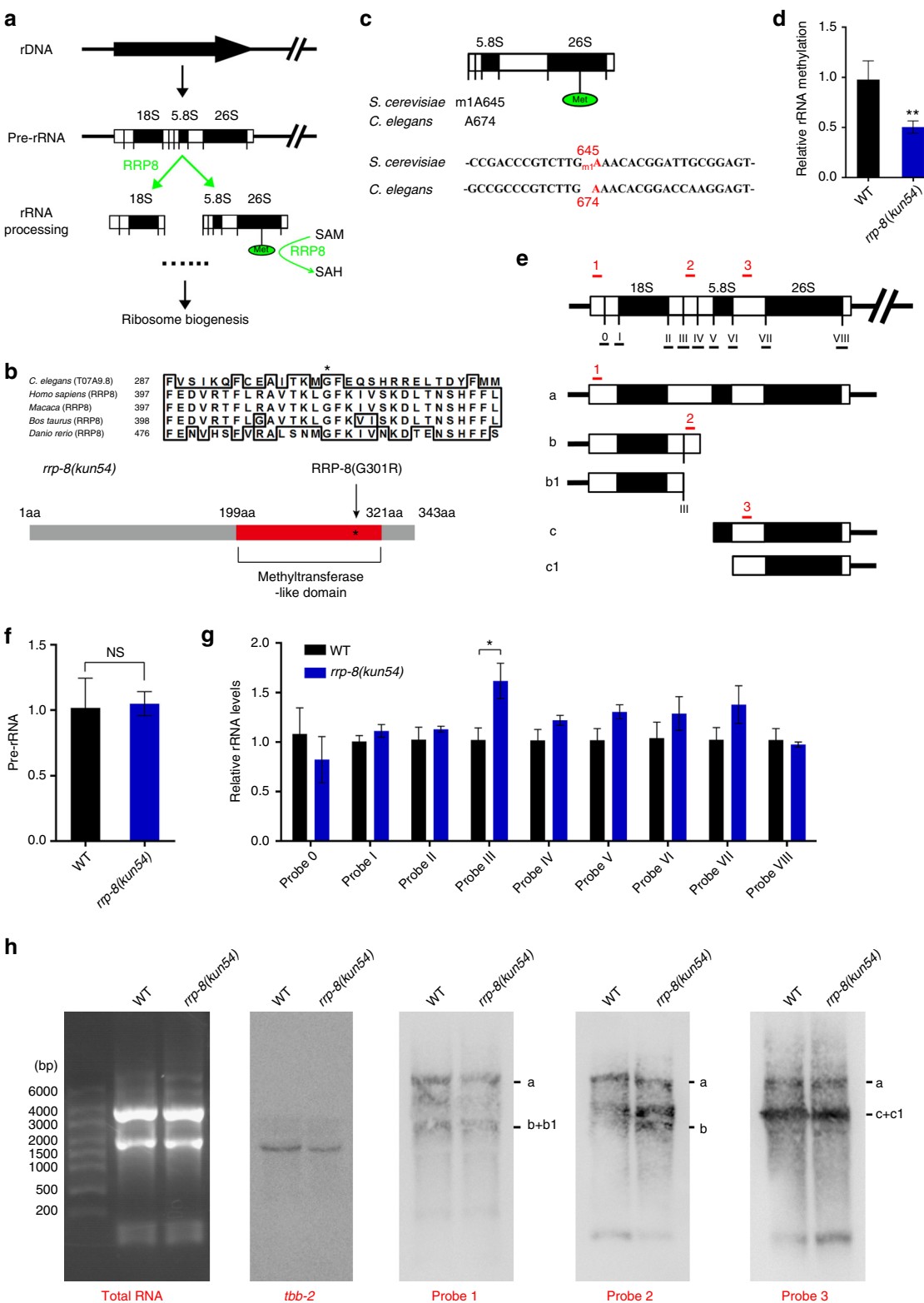

Several studies have reported that EGL-9—a proline hydroxylase, AAK-2—one of two *C. elegans* homologs of the catalytic α-subunit of AMP-activated protein kinases, and PHA-4—a FoxA transcription factor, mediate the extended lifespan of the *rsks-1* (*ok1255*) mutant[36,37]. Therefore, we determined whether nucleolar stress-induced lipid accumulation might be dependent on the activity of these factors. However, the increased lipid accumulation of the *rrp-8*(*kun54*) mutant was unchanged in either the *egl-9* (*sa307*) mutant or *aak-2*(*ok524*) mutant backgrounds (Supplementary Fig. 5A and B), suggesting that nucleolar stress-induced lipid accumulation is independent on both factors.

The temperature-sensitive strain *pha-4*(*zu225*);*smg-1*(*cc546ts*) has been reported to show inactivation of PHA-4 at a restrictive temperature of 20 °C, due to the mutation (*cc546ts*) background of *smg-1* that encodes a component of nonsense-mediated decay (NMD) pathway[38]. Remarkably, the *pha-4*(*zu225*) mutation successfully abolished the increased lipid accumulation in *rrp-8* (*kun54*), *rsks-1*(*ok1255*), and AD-treated worms (Fig. 4a–d), in which the TAG content, quantitated by TLC/GC analysis (Fig. 4d), and LD size (Fig. 4b) were similar to those of the *pha-4*(*zu225*) mutant, whereas the *smg-1*(*cc546ts*) mutation alone, as a control, had no effect (Fig. 4c). PHA-4 has been reported to play a critical role in pharyngeal development[39], raising a question whether the suppression of nucleolar stress-induced lipid accumulation in the *pha-4*(*zu225*) mutant might be due to its pharyngeal pumping defect. To rule out this possibility, we examined *eat-2*, which encodes a nicotinic acetylcholine receptor subunit that functions in the pharyngeal muscle, the mutation of which led to a feeding defect with reduced pharyngeal pumping[40,41]. Although *eat-2*(*ad465*) mutant worms showed obviously decreased lipid accumulation compared with WT worms, their TAG content and LD size still increased in the *rrp-8*(*kun54*) mutant background (Supplementary Fig. 6A–C). Thus, the suppression of nucleolar stress-induced lipid accumulation is probably not a result of a feeding defect in in *pha-4*(*zu225*) mutant, even though the phenotype of *pha-4* mutant is stronger than *eat-2* mutant. Importantly, the messenger RNA level of *pha-4* (Fig. 4e) and translational expression of PHA-4 indicated by the fluorescence of PHA-4::GFP {*wgIs37[pha-4::GFP]*} (Fig. 4f and g) and western blotting with the anti-GFP antibody (Fig. 4h) were obviously induced in *rrp-8*(*kun54*), *rsks-1*(*ok1255*), and AD-treated worms. These results suggest that PHA-4 is a nucleolar stress sensor that is required for nucleolar stress-induced lipid accumulation.

**RPL-11.2/RPL-5 mediate nucleolar stress response**. Next, we examined the factors required for the upregulation of PHA-4 by nucleolar stress. As reported previously, p53-dependent or independent nucleolar stress pathways are almost all mediated by RPs RPL11 and RPL5, which may function in gene regulation[12,42]. RPL11/RPL5 are necessary for nucleolar stress pathways, as depletion of RPL11/RPL5 impairs the p53-dependent or independent nucleolar stress response[6,43]. *rpl-11.1* and *rpl-11.2* are two orthologs of mammalian *rpl11*, and *rpl-5* is an ortholog of mammalian *rpl5* in *C. elegans*. RNA interference (RNAi) knockdown of either *rpl-11.2* or *rpl-5* clearly abolished the increased expression of PHA-4::GFP in *rrp-8*(*kun54*) worms in comparison with that in WT worms (Fig. 5a and b). In contrast, RNAi knockdown of *rpl-11.1*, which showed a high RNAi efficiency (Fig. 5c), had no effect on PHA-4::GFP expression. Furthermore, RNAi reduction of *rpl-11.2* or *rpl-5* apparently suppressed the increased lipid accumulation, as indicated by Nile Red staining of fixed worms (Fig. 5d) and quantification by TLC/GC (Fig. 5e), as well as enlarged LDs (Fig. 5f), in both *rrp-8* (*kun54*) and *rsks-1*(*ok1255*) mutant worms. Taken together, these data clearly indicate that the RPs RPL-11.2 and RPL-5, consistent with their mammalian orthologs, have conserved functions to mediate the upregulation of PHA-4 expression and lipid accumulation during nucleolar stress.

**PHA-4 transactivates the expression of lipogenic genes**. As nucleolar stress-induced lipid accumulation is dependent on the transcription factor PHA-4, we subsequently performed high-throughput RNA-seq to explore the downstream genes involved in lipid accumulation. A comparison of the transcriptome profiles of WT and *rrp-8*(*kun54*) mutant worms revealed that the expression of *pha-4* was consistently elevated (4.8-fold). Interestingly, the transcription of many lipid metabolic genes differed between WT and the *rrp-8*(*kun54*) mutant (Fig. 6a). In particular, transcriptional expression of the lipogenic genes *fasn-1*, *pod-2*, and *dgat-2* was increased in the *rrp-8*(*kun54*) mutant compared with WT worms (Fig. 6a). *pod-2* encodes acetyl-CoA carboxylase and *fasn-1* encodes fatty acid synthase; both genes are involved in the de novo biosynthesis of fatty acids, whereas *dgat-2* encodes diacylglycerol *O*-acyltransferase 2, which incorporates fatty acids into diacylglycerol (DAG) to biosynthesize TAG. Consistently, QPCR analysis further confirmed that the mRNA expression of *fasn-1*, *pod-2*, and *dgat-2* was indeed increased in *rrp-8*(*kun54*), *rsks-1*(*ok1255*), and AD-treated worms compared with WT worms (Fig. 6b). However, *fasn-1*, *pod-2*, and *dgat-2* expression was suppressed in the *pha-4*(*zu225*) mutant background (Fig. 6b), suggesting that the upregulated expression of these genes requires the activity of PHA-4 under nucleolar stress.

Next, we investigated whether the transcription factor PHA-4 could directly bind to and transactivate the expression of the above lipogenic genes. We analyzed the potential binding sites of

**Fig. 2** Mutation of RRP-8 impairs rRNA methylation and processing. **a** Schematic diagram of ribosome biogenesis in yeast and mammals. RRP8 may be involved in pre-rRNA cleavage and methylation. **b** Top panel: alignment of partial amino acid sequences of RRP-8 across several organisms, including *Homo sapiens* (NP_056139.1), *Macaca mulatta* (AFH30501.1), *Bos taurus* (NP_001179303.1), and *Danio rerio* (XP_017206635); the conserved glycine is marked by an asterisk. Bottom panel: schematic diagram of the mutation (G301R) in the methyltransferase-like domain (MLD) of RRP-8 and the mutation site is indicated by an asterisk. **c** The m1A modification site of 25/26S rRNA in yeast and *C. elegans*. **d** The rRNA methylation rate at the indicated site (A674) in 26S rRNA. **e** Schematic diagram of the pre-rRNA (a) and its processed intermediates (b, b1, c, and c1) in *C. elegans*[14, 22]. The cleavage sites (0–VIII) are indicated in the pre-rRNA. The underlined "_" under 0–VIII represents QPCR regions for detection of the splicing efficiency. The positions of probes 1–3 used for northern blot analysis are shown in the indicated regions (red "_"). **f, g** Detection of Pre-rRNA levels (**f**) and splicing efficiency of probable cleavage sites in pre-rRNA (**g**) by RT-qPCR. The levels of rRNA at each indicated site (0–VIII) were detected in 1-day-old WT and *rrp-8*(*kun54*) worms by RT-qPCR with the corresponding primers listed in the corresponding method. Data are presented as the mean $2^{-ddCt} \pm$ SD of four biological replicates using *tbb-2* as a reference gene. **h** Northern blot analysis of pre-rRNA and its processed intermediates in WT and *rrp-8*(*kun54*) worms. Equal amounts of total RNA (5 μg) were loaded on a 1.2% denaturing formaldehyde/agarose gel and analyzed by northern blotting with digoxin-labeled probes 1–3 indicated in **e**. *tbb-2* was used as a control of RNA quality. Data are presented as the mean ± SD of four biological repeats. Significant difference between WT and the *rrp-8*(*kun54*) mutant, Student's *t*-test, **$P < 0.01$, *$P < 0.05$. NS, no significant difference

their promoter regions by combining modENCODE PHA-4::GFP ChIP-seq analyses and characterized PHA-4-responsive elements in *C. elegans*[44,45]. Several PHA-4-responsive elements were identified in the promoter regions of the *fasn-1*, *pod-2*, and *dgat-2* genes (Fig. 6c). *taf-1* encoding an ortholog of human TATA-binding protein associated factor TAF1L (TAFII250) and

*myo-2* encoding a muscle-type-specific myosin heavy chain isoform were previously reported to be the negative and positive control of PHA-4::GFP ChIP, respectively[46]. Furthermore, chromatin immunoprecipitation (ChIP)-QPCR analysis revealed that PHA-4 had the ability to bind to the promoter regions of *fasn-1*, *pod-2*, and *dgat-2* (Supplementary Fig. 7A), and the

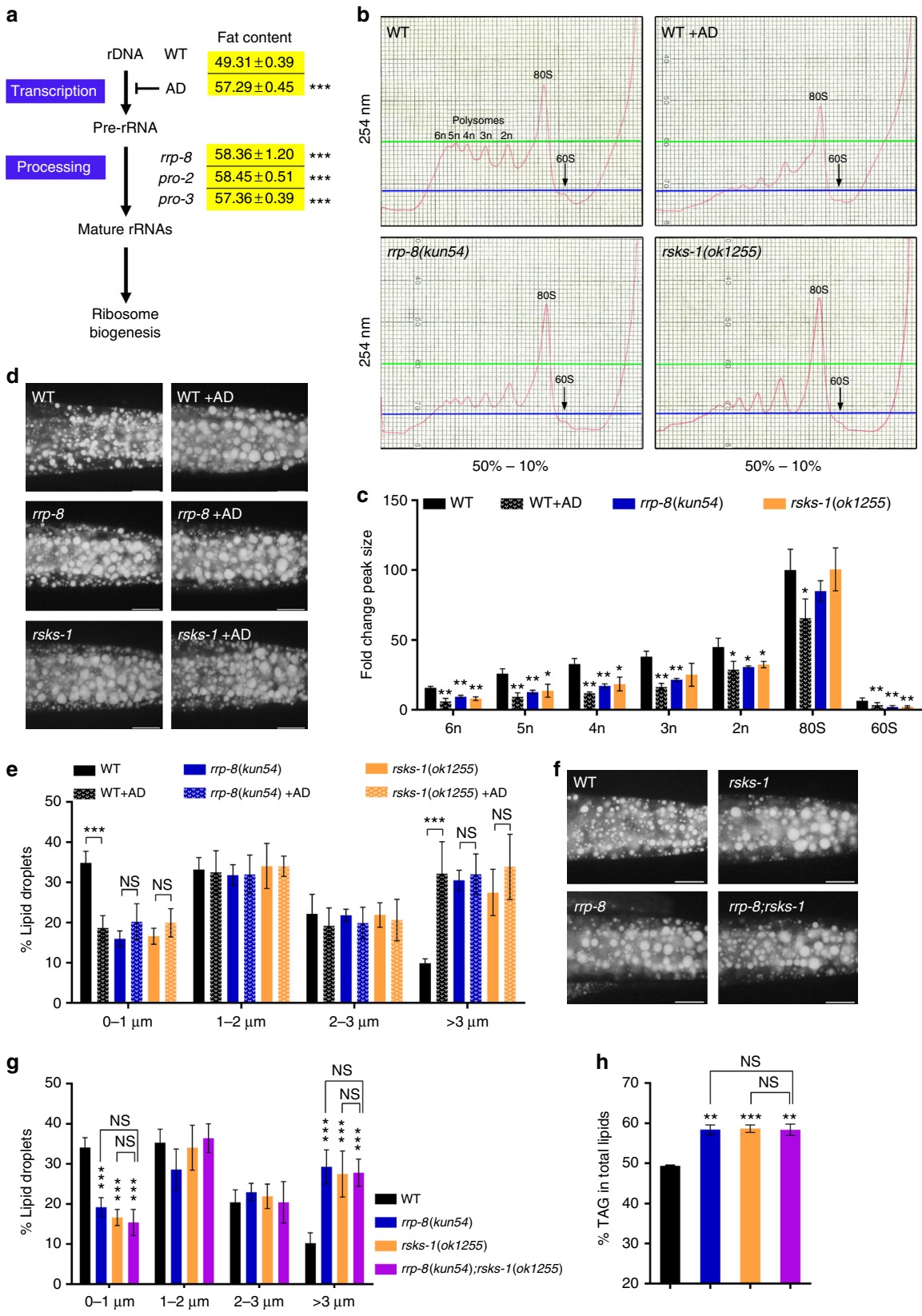

binding of PHA-4 to these genes showed an increased tendency in both the *rrp-8(kun54)* and *rsks-1(ok1255)* mutants compared with WT worms (Fig. 6d–f). Altogether, these data suggest that the transcription factor PHA-4 responds to nucleolar stress, enhancing its binding and consequent transactivation of the expression of *fasn-1*, *pod-2*, and *dgat-2*.

To examine whether the binding of PHA-4 to the promoters of lipogenic genes was necessary for their upregulation under nucleolar stress, we opted to investigate the expression of *dgat-2*, as it is involved in the last step of TAG biosynthesis and its promoter has the minimum number of putative binding sites for PHA-4 in comparison with *fasn-1* and *pod-2* (Fig. 6c). Three PHA-4-binding sites among which two were present in the *m1* region (blue circles, from − 2000 to − 1762) and one in the *m2* region (green rectangle, from − 1123 to − 624) were found in the promoter region of *dgat-2* (Fig. 6c and Supplementary Fig. 7B). We generated four constructs with the full-length promoter (*full*) or truncated promoters (*m1*, *m2*, *m1* + *m2*) of *dgat-2* fused to GFP as a reporter (Supplementary Fig. 7B). The expression of GFP with either full-length promoter [*Pdgat-2(full)::GFP*] or *m1* truncated promoter [*Pdgat-2(m1)::GFP*] was upregulated in response to AD treatment compared with the control (Supplementary Fig. 7C and D). However, loss of the *m2*-binding region in the *dgat-2* promoter [*Pdgat-2(m2)::GFP*] and [*Pdgat-2 (m1 + m2)::GFP*] resulted in not only decreased expression of DGAT-2::GFP compared with the full-length *dgat-2* promoter [*Pdgat-2(full)::GFP*] but also abolished the response to AD treatment (Supplementary Fig. 7C and D). Taken together, these results clearly indicate that the *m2*-, but not the *m1*-, binding site of *dgat-2* promoter is crucial for PHA-4 binding to transactivate its expression.

**Lipogenic genes account for increased lipid accumulation**. As nucleolar stress induces transcriptional expression of the lipogenic genes *fasn-1*, *pod-2*, and *dgat-2*, we subsequently examined whether nucleolar stress-induced lipid accumulation depends on the functions of these genes. RNAi knockdown of *fasn-1* and *pod-2* from synchronized eggs resulted in severe developmental arrest at the L1 stage in N2, *rrp-8(kun54)*, and *rsks-1(ok1255)* worms (Supplementary Fig. 8A), suggesting a vital role for both genes in normal life activities. Moreover, RNAi knockdown of *fasn-1* and *pod-2* from the L2/L3 developmental stage resulted in low lipid accumulation, as indicated by Nile Red staining of fixed worms and a reduced LD size in WT, and *rrp-8(kun54)* and *rsks-1 (ok1255)* mutants (Supplementary Fig. 8B and C), suggesting that the elevated lipid accumulation induced by nucleolar stress triggered by mutations in *rrp-8(kun54)* and *rsks-1(ok1255)* was dependent on the activities of FASN-1 and POD-2.

DGAT-2 catalyzes the conjugation of a fatty acyl-CoA to DAG, to form TAG. Previous studies in *C. elegans*[47] and mammalian systems[48] have shown that DGAT-2 is a LD protein that is required for LD expansion. To verify the upregulated

transcriptional expression of *dgat-2* under nucleolar stress, we generated an integrated translational strain of DGAT-2::GFP {*kunSi148[Pdgat-2::dgat-2::gfp]*} driven by its own promoter. Consistently, the fluorescence intensity of DGAT-2::GFP (Fig. 7a and b) and immunoblotting with an anti-GFP antibody (Fig. 7c) were apparently increased in *rrp-8(kun54)*, *rsks-1(ok1255)*, and AD-treated worms. By contrast, expression of DGAT-2::GFP {*hjSi56[Pvha-6::dgat-2::GFP]*}, which is driven by the promoter of *vha-6* gene that encodes an ortholog of subunit a of V-ATPase, was unchanged in these worms (Supplementary Fig. 9), suggesting that the increased expression of DGAT-2 reflected transcriptional upregulation. Importantly, the increased fluorescence expression of DGAT-2:: GFP in *rrp-8(kun54)*, *rsks-1(ok1255)*, and AD-treated worms was remarkably abrogated following RNAi knockdown of *pha-4* (Fig. 7d and e). These results suggest that the upregulation of *dgat-2* under nucleolar stress is dependent on the transactivity of PHA-4.

Next, to examine whether nucleolar stress-induced lipid accumulation was also dependent on DGAT-2 activity, we generated two knockout alleles (*kun140* and *kun141*, Supplementary Table 2) of *dgat-2* using CRISPR/cas-9 technology (Fig. 7f). As expected, both the *kun140* and *kun141* mutations of *dgat-2* successfully suppressed the increased lipid accumulation, as indicated by Nile Red staining of fixed worms and quantification by TLC/GC analysis, and an enlarged LD size in *rrp-8(kun54)* and *rsks-1(ok1255)* mutant worms (Fig. 7g–i). These results suggest that DGAT-2 activity is necessary for nucleolar stress-induced lipid accumulation.

**Nucleolar stress promotes worm survival during starvation**. To survive under prolonged periods of starvation, animals often shift their metabolism to increased nutrient storage and efficiency of energy utilization. As nucleolar stress promotes lipid accumulation, we examined whether this phenomenon could affect starvation survival in *C. elegans*. Synchronized L4 worms were subjected to M9 buffer without food to examine their survival. Indeed, *rrp-8(kun54)* and *rsks-1(ok1255)* mutant worms (Supplementary Fig. 10A and Supplementary Table 1), as well as AD-treated worms (Supplementary Fig. 10B and Supplementary Table 1), apparently lived longer than WT worms under starvation conditions. Compared with WT worms, *pha-4(zu225)* mutant worms died quickly at the early stage of starvation (Supplementary Fig. 10B–F and Supplementary Table 1). The extended starvation survival of *rrp-8(kun54)*, *rsks-1(ok1255)* and AD-treated worms was reversed to some degree in the *pha-4 (zu225)* mutant background (Supplementary Fig. 10C, D and Supplementary Table 1). Far more importantly, a similar effect was shown in the *dgat-2(kun140)* mutant (Supplementary Fig. 10E, F and Supplementary Table 1), in which the increased lipid accumulation by nucleolar stress was suppressed (Fig. 7g–i),

---

**Fig. 3** Nucleolar stress triggered by the perturbation of ribosome biogenesis leads to excessive lipid accumulation. **a** The general processes of ribosome biogenesis. The fat content percentage of triacylglycerol (% TAG) in total lipids (TAG + phospholipids, PL) of each worm strain was quantitated using TLC/GC analysis. Data are presented as the means ± SD of three biological repeats. **b** Ribosomal profiles of WT, *rrp-8(kun54)*, *rsks-1(ok1255)*, and AD-treated worms. **c** Quantification of the polysomes. The areas under each peak were calculated using ImageJ software. The fold-change of each peak size for a specific worm strain is presented compared to the 80S peak size of wild type (WT). Data are presented as the means ± SD of three biological repeats. **d**, **f** Nile Red staining of fixed worms. Representative animals; the anterior is indicated on the left, and the posterior is indicated on the right. Scale bar represents 20 μm. **e**, **g** Distribution of the lipid droplet size (% lipid droplets), as measured by Nile Red staining of fixed worms from **d** and **f**, respectively. Data are presented as the means ± SD of 10 animals for each worm strain. **h** Percentage of triacylglycerol (% TAG) in total lipids (TAG + phospholipids, PL) analyzed by TLC/GC. Data are presented as the means ± SD of four biological repeats. Significant difference between a specific worm strain and its genetic background strain, Student's *t*-test, ***P < 0.001, **P < 0.01, *P < 0.05. NS, no significant difference

indicating that starvation survival actually requires excessive lipid storage.

## Discussion

Ribosome biogenesis in the nucleolus is an extraordinarily energy-consuming process. The present study revealed that nucleolar stress elicited by gene mutations in *rsks-1*, *rrp-8*, *pro-2*, and *pro-3* participates in pre-rRNA processing and the AD treatment blocks pre-rRNA transcription, consistently leading to excessive lipid accumulation in *C. elegans*. Thus, our results systematically demonstrate that nucleolar stress alters the energy

status to increase lipid accumulation in an intact animal model in *C. elegans*.

The tumor suppressor p53 is considered to be an essential factor to monitor the integrity of ribosome biogenesis[6,7]. Consistently, we observed that the expression of CEP-1, an ortholog of human p53 in *C. elegans*, was actually elevated in the *rrp-8*(*kun54*) mutant and AD-treated worms (Supplementary Fig. 3A and B), indicating an evolutionarily conserved role for p53 as a sensor of nucleolar stress. However, inactivation of CEP-1 had no effect on lipid accumulation and failed to repress the excessive lipid accumulation triggered by nucleolar stress

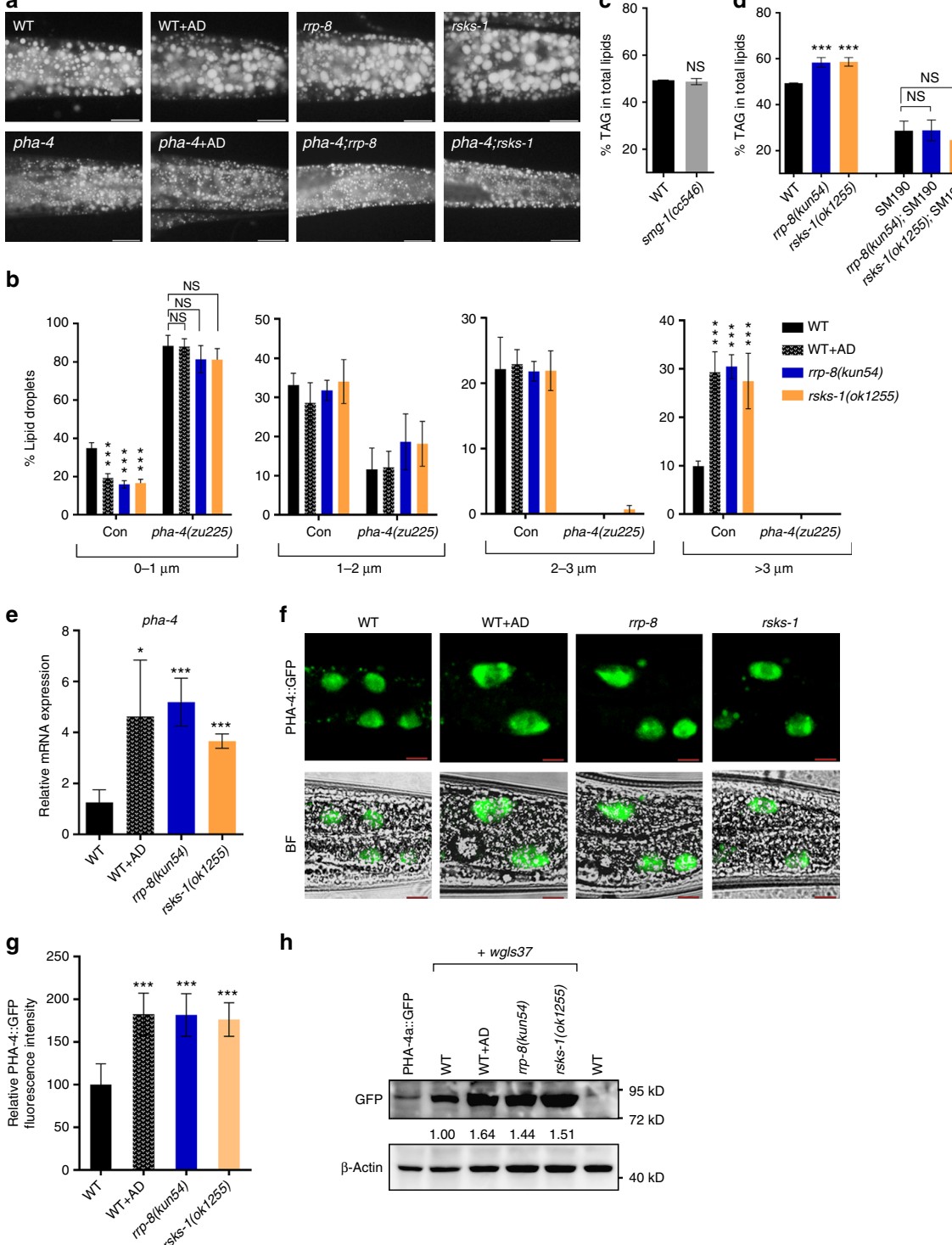

(Supplementary Fig. 3C and D) in worms, suggesting a novel sensor existed that links nucleolar stress to lipid accumulation in *C. elegans*.

By contrast, the results of the present study uncovered that the transcription factor PHA-4 plays a central role in response to nucleolar stress and lipid accumulation. First, ribosome biogenesis in eukaryotic cells occurs in the nucleolus. Unlike expression of CEP-1::GFP mainly in germline cells (Supplementary Fig. 3A), PHA-4::GFP is also expressed in the nucleolus of intestine cells (Fig. 4e), the major sites of fat synthesis and storage in *C. elegans*. This finding may imply a direct function of PHA-4 in the nucleus of intestine cells to sense nucleolar stress and subsequently initiate lipid accumulation. Second, both mRNA and protein levels of the transcription factor PHA-4 are induced in response to nucleolar stress triggered by both gene mutations and AD treatment (Fig. 4d–f). Third, nucleolar stress promotes the binding of PHA-4 to the promoters of the lipogenic genes *fasn-1*, *pod-2*, and *dgat-2*, and their induced expression in response to nucleolar stress is dependent on the activity of PHA-4. In particular, the PHA-4-binding site *m2* of *dgat-2* is critical for its transcription. Although the transcription factors SBP-1/SREBP[31,32] and DAF-16/FOXO[49] are master regulators of lipogenesis, these proteins do not respond to nucleolar stress (Supplementary Fig. 4A and B). Ultimately, inactivation of *pha-4* and its targets, *fasn-1*, *pod-2*, and *dgat-2*, successfully suppresses nucleolar stress-induced lipid accumulation. Thus, we revealed a distinct PHA-4-dependent lipogenesis pathway that concurrently associates nucleolar stress with lipid accumulation. We propose that nucleolar stress activates the transcription factor PHA-4, which is mediated by RPs RPL-11.2/RPL-5. Subsequently, PHA-4 binds to and transactivates the expression of lipogenic genes to promote fat biosynthesis and accumulation (Fig. 8). The family of Forkhead proteins is present in almost all eukaryotes[50], and the mammalian FoxA family, comprising FoxA1, FoxA2, and FoxA3, contains critical regulators of mammalian development and metabolism[51]. Similarly, our results uncover that PHA-4 is also a critical regulator of lipogenesis in addition to being a pioneer transcription factor to recruit RNA polymerase II[46] and a critical regulator of development and longevity[40,44]. It would be interesting to investigate whether PHA-4/FoxA acts as an evolutionarily conserved sensor of nucleolar stress across eukaryotes.

In eukaryotes, in addition to their roles as components of translation machinery, RPs have a variety of extra-ribosomal functions[52]. The RPs RPL11 and RPL5 have been showed to mediate almost all p53-dependent or independent nucleolar stress pathways in mammals[12,42]. Remarkably, our present work consistently reveals that nucleolar stress-induced upregulation of PHA-4 and lipid accumulation also depend on the mammalian orthologs RPL-11.2/RPL-5 (Fig. 5), suggesting that the extra-ribosomal function of RPL11/RPL5 in response to nucleolar stress is evolutionarily conserved in *C. elegans* and mammals. Taken together, we speculate that the RPs RPL11 and RPL5 may have a central role that response to various upstream input-caused nucleolar stresses, and then pass to PHA-4, P53/CEP-1, or other factors to transactivate their corresponding downstream targets, eventually leading to distinct physiological outputs, such as lipid accumulation, cell proliferation, and differentiation, and so on.

LDs are intracellular organelles that dynamically respond to the metabolic states of cells and tissues to regulate lipid accumulation[53]. Several lipid metabolic genes are involved in the growth and expansion of LDs. We observed that nucleolar stress activated the expression of the lipogenic genes *fasn-1*, *pod-2*, and *dgat-2*, and that inactivation of *fasn-1*, *pod-2*, and *dgat-2* abolished nucleolar stress-induced lipid accumulation. In addition, we previously showed that another key lipogenic enzyme, stearoyl-CoA desaturase, regulates the LD size and fat storage of *rsks-1* mutants[34]. Therefore, activated lipogenesis primarily accounts for nucleolar stress-induced lipid accumulation, although we could not exclude the possibility that other lipid metabolic genes might also contribute to nucleolar stress-induced lipid accumulation, as their expression was also altered in the transcriptome profile (Fig. 6a).

Strikingly, nucleolar stress-induced excessive lipid accumulation is beneficial for worms to extend survival under starvation, which depends on the function of PHA-4 and its direct target DGAT-2. In support of this observation, a very recent report also showed that reduced function of cytoplasmic aminoacyl transfer RNA synthetases (ARS genes) leads to increased fat storage and extended starvation survival in *C. elegans*[54]. In contrast, genetic activation of protein synthesis results in opposite effects in *Drosophila*[55]. Therefore, these lines of results consistently demonstrate that metabolic alteration via reduced ribosome biogenesis or protein production may be a useful strategy for animals to survive under prolonged periods of starvation.

In humans, the dysfunction of ribosome biogenesis often leads to disorders known as ribosomopathies or lethalities[56,57]. Among these disorders, Prader–Willi syndrome (PWS) is a complex disease that is characterized by obesity, neonatal hypotonia, short stature, and other clinical syndromes[58]. PWS patients suffer from a loss of the HBII-85 box C/D small nucleolar RNA cluster, which catalyzes 2′-*O*-ribose methylation of rRNA, participates in pre-rRNA folding, and is also essential for pre-rRNA processing[59]. The high fat phenotype of ribosome biogenesis gene mutants in *C. elegans* may somewhat mimic the obesity symptoms of PWS. It will be interesting to explore the role of the RPL-11/RPL5-PHA-

**Fig. 4** PHA-4 responses to nucleolar stress. **a** Nile Red staining of fixed worms. Synchronized *smg-1(cc546);pha-4(zu225)* eggs were hatched in M9 buffer to L1 worms at 24 °C. Subsequently, L1 worms were transferred to NGM plates, raised at 20 °C, and then collected when they reached 1 day of adulthood for analysis of lipid contents. Representative animals; the anterior is indicated on the left and the posterior is indicated on the right. Scale bar represents 20 μm. **b** Distribution of the lipid droplet size (% lipid droplets) as measured by Nile Red staining of fixed worms from **a**. Data are presented as the means ± SD of 10 animals for each worm strain. Significant difference between WT and a specific worm strain, ANOVA, ***$P < 0.001$. NS, no significant difference. **c**, **d** Percentage of triacylglycerol (% TAG) in total lipids (TAG + phospholipids, PL) analyzed by TLC/GC. Data are presented as the means ± SD of four biological repeats. Significant difference between WT and a specific worm strain, Student's *t*-test, ***$P < 0.001$. NS, no significant difference. **e** Relative mRNA expression of *pha-4* in WT and mutant worms. Data are presented as the means ± SD of three biological repeats. Significant difference between WT and a specific worm strain, ANOVA, ***$P < 0.001$, *$P < 0.05$. **f** Confocal microscopy of PHA-4::GFP {*wgIs37[pha-4::GFP]*} in WT, *rrp-8(kun54)*, *rsks-1* (*ok1255*), and AD-treated worms at 1 day of adulthood. BF, bright field. Representative animals; the anterior is indicated on the left and the posterior is indicated on the right. Scale bar represents 10 μm. **g** Quantification of the fluorescence intensity of PHA-4::GFP from **f**. Data are presented as the means ± SD of at least 20 worms for each worm strain. Significant difference between WT and *rrp-8(kun54)* mutant background, ANOVA, ***$P < 0.001$. **h** Immunoblotting of PHA-4::GFP {*wgIs37[pha-4::GFP]*} with anti-GFP antibody in WT, *rrp-8(kun54)*, *rsks-1(ok1255)*, and AD-treated background worms at 1 day of adulthood. WT worms without GFP was used as a negative control and WT worms with PHA-4a::GFP {*kunEx136[Pvha-6::pha-4a::GFP]*} as a positive control. The relative protein levels of PHA-4::GFP were labeled

4-mediated lipogenesis pathway in PWS and other ribosomopathies.

## Methods

**C. elegans strain, RNAi, and primers**. *C. elegans* were maintained on nematode growth media (NGM) with *Escherichia coli OP50* under standard culture conditions. RNAi was performed using the feeding method[60]. RNAi bacteria strains were seeded on NGM supplemented with 100 µg/ml ampicillin and 1 mM IPTG (isopropyl β-D-1-thiogalactopyranoside). On the next day, synchronized L1-stage worms were transferred to prepared NGM plates for RNAi treatment. Young adult worms were collected for further analyses after 48 h, unless specifically indicated. The WT strain was Bristol N2. The complete worm strains, including the transgenic strains used in the present study, are listed in Supplementary Table 2. The sequence information of primers used in this study was listed in Supplementary Table 3.

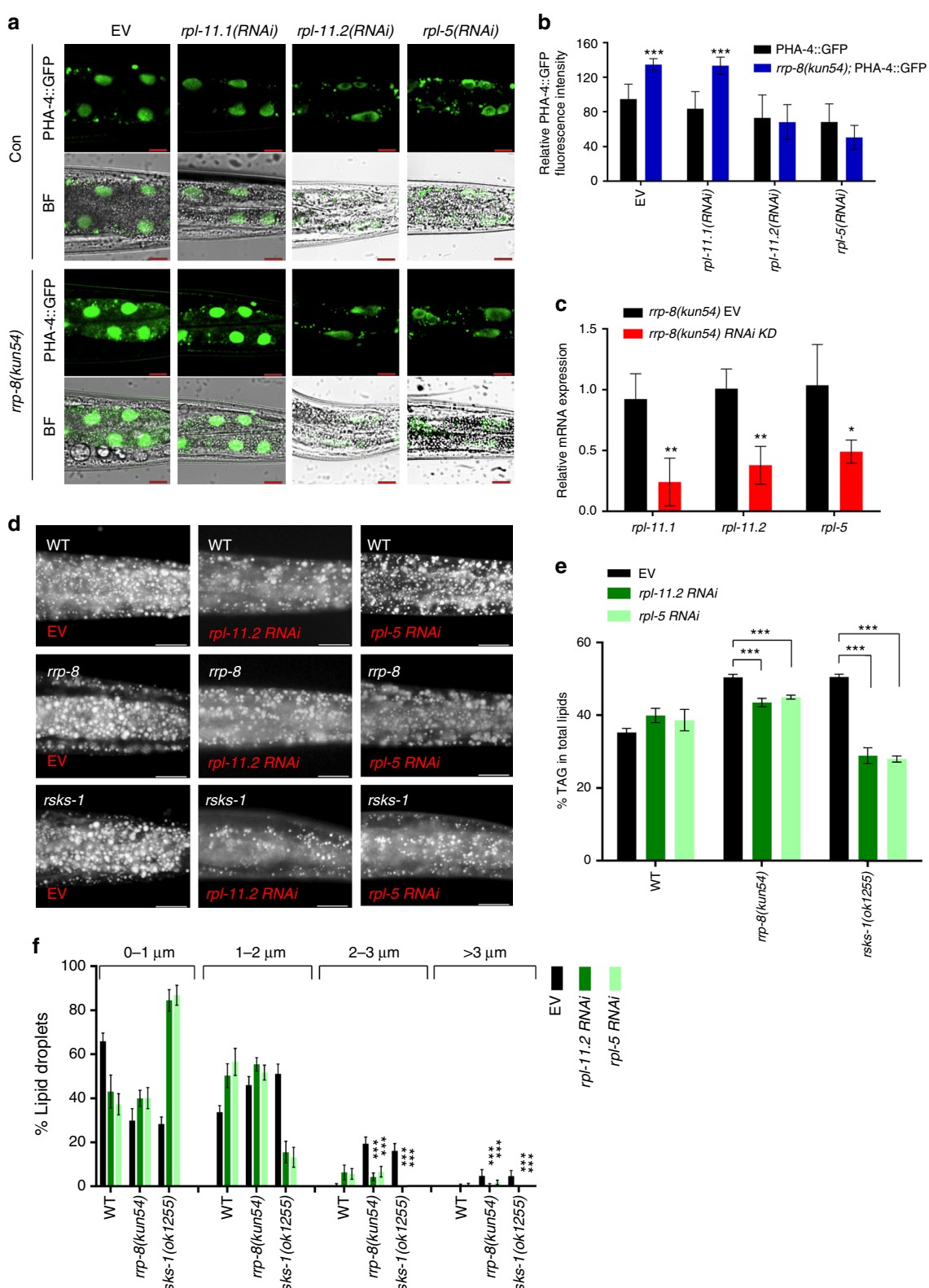

**Isolation and identification of kun54.** The kun54 mutation was isolated from a forward genetic screen of fat regulators by EMS according to a report by Jorgensen and Mango[61]. In general, we performed the EMS screen using WT L4 animals (F1) treated with 50 mM EMS. The progeny from the single F2 generation were fixed and stained using Nile Red to screen mutants with various LD sizes and lipid accumulation[15]. The mutation of kun54 was mapped using a SNP method according to Davis et al.[17] and identified using whole genome sequencing according to Sarin et al.[18].

**Construction of transgenic strains.** The transgenes were generated by micro-injection[62]. In general, DNA fragments of specific genes and related promoters were amplified using PCR. The amplified DNA fragments were subsequently inserted into the transgenic plasmid pPD95_75 or pCFJ151. For extrachromosomal transgenic strains, transgenic plasmids were injected with a fluorescence marker (pCFJ90) into young adult worms. The positive transgenic worms were selected based on fluorescence expression.

For single copy insertion to generate integrated transgene strains, MosI integration was used[62]. The injection mixture, containing 50 ng $\mu l^{-1}$ transgenic plasmid, 50 ng $\mu l^{-1}$ pJL43.1, 5 ng $\mu l^{-1}$ pCFJ104, and 3 ng $\mu l^{-1}$ pCFJ90, was injected into EG4322 [ttTi5605;unc-119(ed3)]. At least 400 transgenic worms with fluorescence and rescued locomotion of EG4322 were screened. The complete integrated transgenes are listed in Supplementary Data 1. The primer sequences are available in Supplementary Data 2.

**CRISPR/cas-9 technology to generate rrp-8 and dgat-2 mutants.** The generation of rrp-8(kun122), dgat-2(kun140), and dgat-2(kun141) mutants using CRISPR/cas-9 technology of dual dingle guide RNA (sgRNA)-directed gene knockout was performed as follows[63,64]. The sgRNA sequences of a specific gene were designed based on the target sequence characterized by G/A(N)19NGG, which was subsequently placed into the pU6::unc-119_sgRNA plasmid by overlap extension PCR. Single- or dual-sgRNA plasmids (50 ng $\mu l^{-1}$) mixed with 50 ng $\mu l^{-1}$ cas-9 expression plasmid and 3 ng $\mu l^{-1}$ pCFJ90 were injected into WT worms to generate point or DNA fragment deletion mutations. The sgRNA sequences are available in Supplementary Data 2.

**Construction of rrp-8(kun54);SM190 and rsks-1(ok1255);SM190.** SM190[pha-4 (zu225);smg-1(cc546ts)] is a worm strain containing two gene mutation, according to a previous report[38]. Temperature sensitive mutant smg-1(cc546ts), which is the component of NMD pathway, exhibits robust NMD activity at 15 °C, but compromises activity at higher temperature (24 °C). At restrictive temperature 15 °C, high NMD activity of SMG-1(cc546) degrades pha-4 mRNA from pha-4(zu225), leading to lethality of SM190 worms at L1 stage, whereas at 24 °C, SMG-1(cc546ts) was compromised and pha-4 mRNA was stabilized and accumulated, leading to survival of SM190 worms. At 20 °C, SM190 presents an intermediate phenotype but lethal nonetheless.

In order to generate rrp-8(kun54);SM190[pha-4(zu225);smg-1(cc546ts)] and rsks-1(ok1255);SM190[pha-4(zu225);smg-1(cc546ts)] mutants, we crossed the rrp-8 (kun54) or rsks-1(ok1255) mutant into the SM190 [pha-4(zu225);smg-1(cc546ts)] background, respectively, at 24 °C. The mutation background of rrp-8(kun54), rsks-1(ok1255), or pha-4(zu225) was identified through PCR and sequencing. In order to identify the homozygous smg-1(cc546ts), 10 lines of each double mutant worms of rrp-8(kun54);pha-4(zu225) or rsks-1(ok1255);pha-4(zu225) that might contain smg-1(cc546ts) background were picked and raised at 24 °C for normal reproduction. Their progenies were then raised at 15 °C to identify the activity of SMG-1(cc546ts). A line of worms, which were lethal at 15 °C, indicated that they contained robust NMD activity, and identified as rrp-8(kun54);SM190[pha-4 (zu225);smg-1(cc546ts)] or rsks-1(ok1255);SM190[pha-4(zu225);smg-1(cc546ts)]. All double and triple mutants that contain SM190[pha-4(zu225);smg-1(cc546ts)] background were raised at 24 °C and their synchronized eggs were hatched to L1 at 24 °C. Next, these synchronized L1 worms were placed onto NGM plates and cultured at 20 °C for experiment analyses.

**Vital dyes staining of LDs.** Oil Red O[65] and LipidTox Red staining[66] of fixed 1-day-old adult worms were performed as follows. In general, ~ 500 one-day-old adult worms were collected and suspended in 1 ml of water on ice, and then 50 μl of freshly prepared 10% paraformaldehyde solution was added and mixed. Worms were immediately frozen in briefly in liquid nitrogen, subjected to two or three incomplete freeze/thaw cycles, and then washed with M9 buffer several times to remove the paraformaldehyde solution. Two microliters of 5 mg $ml^{-1}$ Nile Red was added to the worm pellet and incubated for 30 min at room temperature, with occasional gentle agitation. Worms were washed two or three times with M9 buffer and mounted onto 2% agarose pads for microscopic observation and photography. Images were obtained using identical settings and exposure times with an Olympus BX53 fluorescence microscope (Japan). Images of Nile Red staining were used to measure the diameter of the LDs in the posterior of the intestine with the same area (100 μm × 80 μm) using cellSens Standard software (Olympus, Japan). At least 10 worms were measured for each worm strain.

**Lipids extraction, separation, and analysis.** Worm lipid extraction, separation, and analysis were performed as follows[34]. In general, ~ $4 \times 10^4$ synchronized L1 worms were seeded on NGM plates and grown to young adults before laying egg under standard culture condition. Worms were collected into a glass tube and 7.5 ml ice-cold chloroform/methanol (1:1) was added with vigorous shaking, and extracted at − 20 °C overnight. Then, 3.3 ml Hajra's solution (0.2 M $H_3PO_4$, 1 M KCl) was added to each sample and shaken vigorously and centrifuged at 6000 r.p.m. for 5 min at 4 °C. The lower chloroform phase containing the lipids was removed to a new glass tube, dried under nitrogen, and then re-suspended in chloroform to 200 μl for TLC and GC analyses.

For lipid separation, lipid samples (30 μl for each) were loaded in triplicate on TLC silica plates (Merck) to separate TAG and phospholipids (PLs), and then developed to the top of the plate in the solvent system hexane:diethyl ether:acetic acid (80:20:2). Individual TAG and PL bands were scraped form TLC plates and spiked with a known standard (C15:0), and transesterified (2.5% $H_2SO_4$ in methanol) for GC analysis to determine the relative levels of TAG and PL fractions as previously described[67]. Fatty acids were determined with an Agilent 7890 series gas chromatographer equipped with a 30 × 0.25 mm SP-2380 column (Supelco), with nitrogen as the carrier gas at 1.4 ml/min, and a flame ionization detector. The content of TAG or PL was determined by counting the fatty acids from each and is shown as the percentage of TAG in total lipids (TAG + PL)[34].

**RNA-sequencing and QPCR analysis.** Synchronized 1-day-old adult worms were collected in RNAiso plus (Takara). Total RNA was isolated from each sample for RNA sequencing, northern blotting, and RT-QPCR. RNA-seq was performed after cluster generation; the library preparations were sequenced on an Illumina Hiseq 4000 platform. Complementary DNA was generated from total RNA using the PrimeScript RT reagent kit (catalog number RR047A, Takara Bio, Inc., Japan) with gDNA eraser. mRNA levels were quantified from biological triplicates using SYBR green fluorescence on a real time PCR instrument 7900HT (ABI). Relative abundance (mRNA) was determined using the ΔΔCt method and tbb-2 was used as a reference gene.

**Analysis of the rRNA methylation rate using RT-QPCR.** The levels of rRNA methylation were analyzed by RT-QPCR[68]. A reverse-transcription primer R and a pair of primers (F1 and R1) near A674 on 26S rRNA were designed for QPCR. The sequences of the primers were R: 5′-AGTCACAAGTGACACGGCAC-3′; F1: 5′-ACAGTGTTGCCCATCTCGC-3′; R1: 5′-ACGTCGGCCAATTCGAGAC-3′. The 26S rRNA transcript was specifically reversed transcribed using primer R to

**Fig. 5** RPL-11.2 and RPL5 are required for nucleolar stress induced upregulation of PHA-4. **a** Confocal microscopy of PHA-4::GFP {wgIs37[pha-4::GFP]} in WT and rrp-8(kun54) worms at 1 day of adulthood under treatment of control (empty vector, EV), rpl-11.1, rpl-11.2, and rpl-5 RNAi from L2/L3 larvae stage, respectively. BF, bright field. Representative animals; the anterior is indicated on the left, and the posterior is indicated on the right. Scale bar represents 10 μm. **b** Quantification of the fluorescence intensity of PHA-4::GFP from **a**. Data are presented as the means ± SD of at least 20 worms for each worm strain. Significant difference between WT and rrp-8(kun54) mutant background, Student's t-test, ***P < 0.001. **c** Efficiency of RNAi knockdown (KD). Relative mRNA expression of rpl-11.1, rpl-11.2, and rpl-5 was measured by QPCR in rrp-8(kun54) worms treated with its corresponding RNAi clone from the L2/L3 larvae stage. Data are presented as the means ± SD of three biological repeats. Significant difference between the control (empty vector, EV) and a specific gene of RNAi bacteria, Student's t-test, **P < 0.01, *P < 0.05. **d** Nile Red staining of fixed WT, rrp-8(kun54) and rsks-1(ok1255) worms treated with empty vector (EV), rpl-11.2, and rpl-5 RNAi from the L2/L3 larvae stage. Representative animals; the anterior is indicated on the left and the posterior is indicated on the right. Scale bar represents 20 μm. **e** Percentage of triacylglycerol (% TAG) in total lipids (TAG + phospholipids, PL) analyzed by TLC/GC. Data are presented as the means ± SD of three biological repeats. Significant difference between a specific mutant strain without (EV) and with RNAi knockdown, Student's t-test, ***P < 0.001. **f** Distribution of the lipid droplet size (% lipid droplets) measured by Nile Red staining of fixed worms from **d**. Data are presented as the means ± SD of 10 animals for each worm strain. Significant difference between a specific mutant strain without (EV) and with RNAi knockdown, Student's t-test, ***P < 0.001

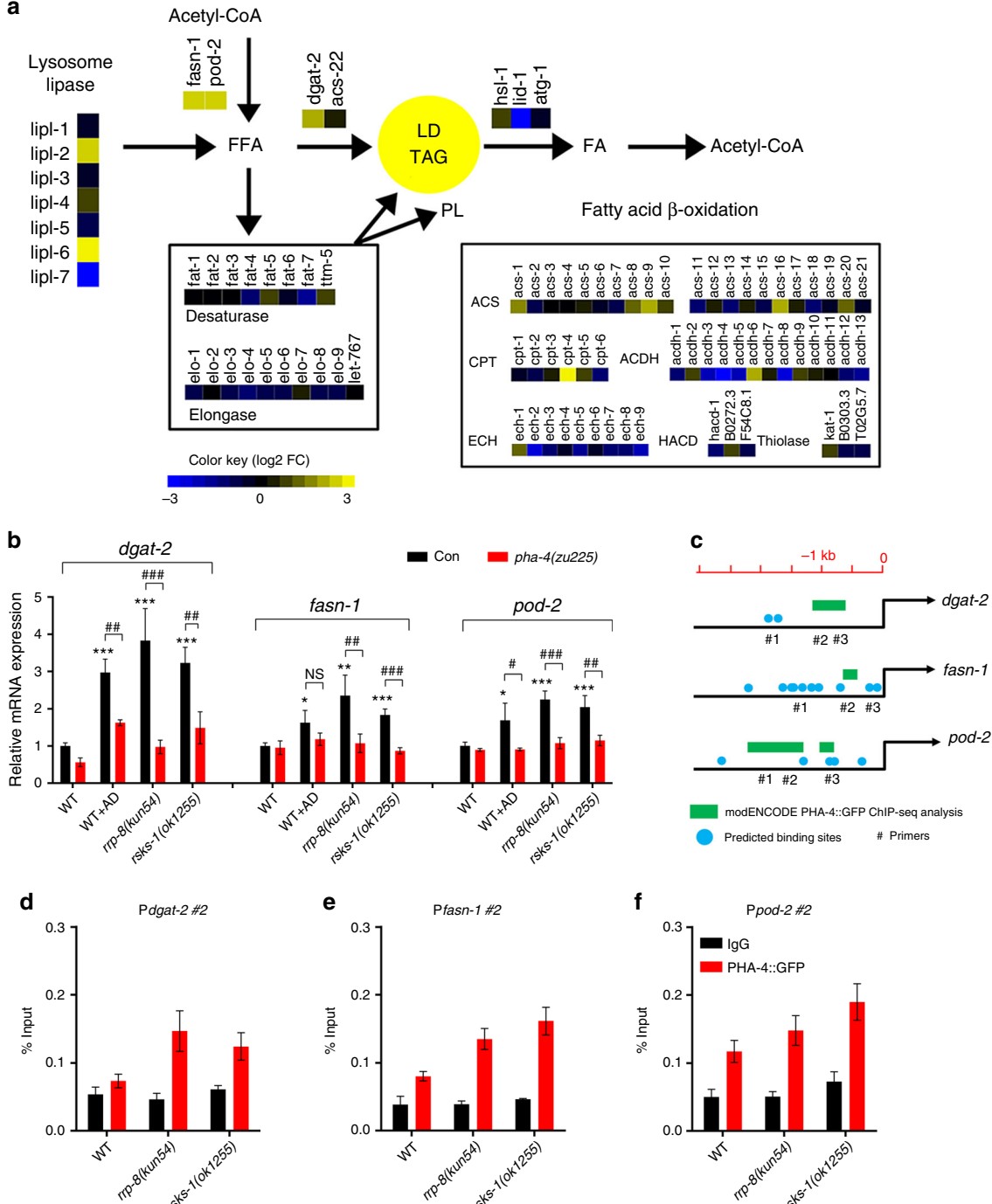

**Fig. 6** Nucleolar stress enhances PHA-4 binding and transactivates the expression of lipogenic genes. **a** Transcriptional expression of lipid metabolic genes using transcriptome profile analysis between WT and *rrp-8(kun54)* mutant worms. Abbreviations: ACDH, acyl-CoA dehydrogenase; ACS, acyl-CoA synthetase; CPT, carnitine palmitoyltransferase; ECH, enoyl-CoA hydratase; FA, fatty acids; HACD, hydroxyacyl-CoA dehydrogenase; PL, phospholipid; TAG, triacylglycerol. **b** Relative mRNA expression of *dgat-2*, *fasn-1*, and *pod-2* in different worm strains or treatments. Data are presented as the means ± SD of four biological repeats. Significant difference between WT and a specific worm strain, ANOVA, ***$P < 0.001$, **$P < 0.01$, *$P < 0.05$. Significant difference between a specific worm strain with and without the *pha-4(zu225)* background, Student's *t*-test, ###$P < 0.001$, ##$P < 0.01$, #$P < 0.05$. NS, no significant difference. **c** Potential PHA-4-binding sites in the promoter region of the lipogenic genes *fasn-1*, *pod-2*, and *dgat-2*. The green rectangles were reported in the modENCODE GFP ChIP-seq analysis and the blue dots were predicted based on the characterized PHA-4-binding elements. #1, #2, and #3 represent different primer pairs in the promoter region of different genes for ChIP-QPCR detection. **d-f** ChIP-QPCR analysis of PHA-4 bound to the promoters of the *fasn-1*, *pod-2*, and *dgat-2* genes. **d** P*dgat-2* #2, **e** P*fasn-1* #2 and **f** P*pod-2* #2 indicate the #2 primer pair in the promoter region of respective genes used for ChIP-QPCR analysis. Data are presented as the means ± SD of three biological repeats

generate cDNA with 100 ng of total RNA extracted from either WT or *rrp-8* (*kun54*) mutant worms and either 10 μM or 1 mM dNTPs in each sample. The reverse transcription system included 200U Hscript Reverse Transcriptase (Vazyme Biotech), 40U RNase inhibitor (Vazyme Biotech), 1 μM specific reverse primer R, and either 10 μM and 1 mM dNTP in each sample. For each reaction, reverse transcription was performed at 25 °C for 5 min, followed by an incubation

at 50 °C for 15 min and at 85 °C for 5 min. RT-QPCR was performed to quantify the specific cDNA levels using the primer pair F1/R1 spanning the methylation site. The relative methylation rate of a specific site on 26S rRNA was calculated as the ratio of the cDNA levels obtained using 1 mM dNTPs to that obtained using 10 μM dNTPs in each sample.

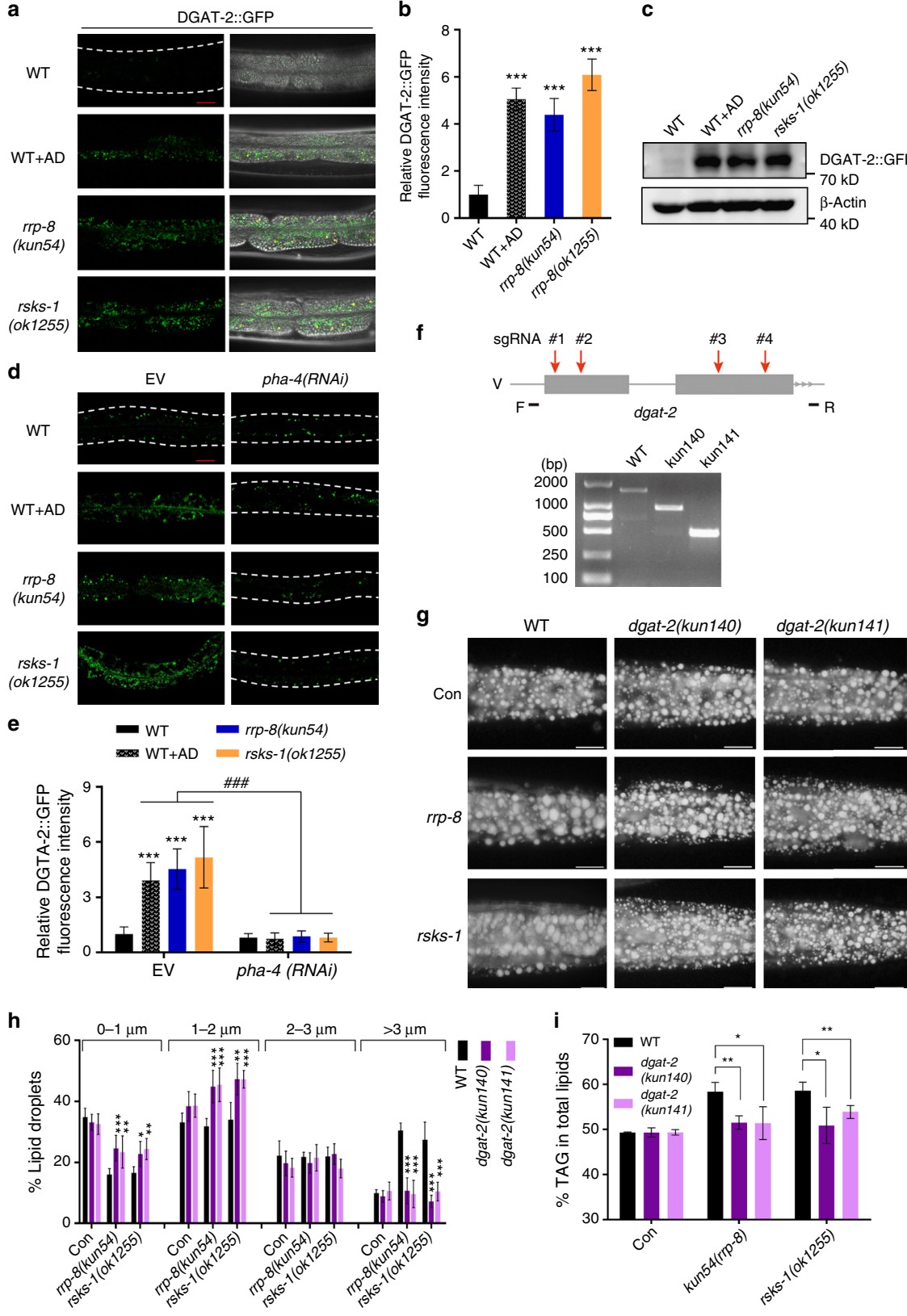

**Splicing efficiency of probable cleavage sites in pre-rRNA**. The pre-rRNA cleavage sites were determined according to previous reports[14,19,22]. Nine pairs of primers spanning various splicing sites were designed for QPCR to detect the cleavage efficiency at these sites: Probe 0 (F: 5′-GAGAAAAACGGTGTCTCGAG-3′; R: 5′-AGACATCACGTCTCAGACC-3′), Probe I (F: 5′-GTGTCCCATCT-CACGATTAG-3′; R: 5′-GTGATATCTGCTCTAATGAG-3′), Probe II (F: 5′-AACGACTTCGTTGTTGCGG-3′; R: 5′-TTCGACACTCAACTGACCG-3′), Probe III (F: 5′-TCAACGTTCCAGTTGAGATG-3′; R: 5′-CGATCATCAA-GACTATCGTC-3′), Probe IV (F: 5′-TGGCTATATGCGTCTAGGC-3′; R: 5′-ATCACCGCATGTCCGTGAAG-3′), Probe V (F: 5′-CTTCACGGA-CATGCCGGTGAT-3′; R: 5′-AGTTGGTGCTATGCGTTCG-3′), Probe VI (F: 5′-CGAACGCATAGCACCAACT-3′; R: 5′-TGTGATGCTTCTGGACTAGG-3′), Probe VII (F: 5′-TCGAATACTGGGGATTCGTC-3′; R: 5′-AGCAGCCAAA-GACTGATCG-3′), and Probe VIII (F: 5′-AGTGAATTCTGCGACGCTTG-3′; R: 5′-TGCAAAGACATGAGTGTAGG-3′). rRNA levels at respective sites (0–VIII) were measured in WT and *rrp-8*(*kun54*) young adult worms and are depicted as the mean $2^{-ddCt} \pm$ SD of four biological replicates. *tbb-2* was used as an internal control.

**AD treatment**. AD was added to NGM medium at a final concentration of 15 ng μL$^{-1}$. Synchronized L1 worms were placed onto the NGM plates supplemented with or without AD, and raised to 1 day of adulthood under standard culture condition, and then collected for further analysis, unless specifically indicated.

**Northern blotting**. Total RNA (5 μg) isolated from either WT or *rrp-8*(*kun54*) was loaded on a 1.2% denaturing formaldehyde/agarose gel for electrophoresis in MOPS (3-*N*-Morpholino propansulfonic acid) buffer. Separated RNA was transferred onto a nylon membrane and hybridized with digoxin (DIG)-labeled anti-sense RNA probes (1–3). DIG-labeled RNA probes were generated using a DIG northern starter kit (Roche). DNA templates with the T7 promoter for transcription were prepared by PCR from cDNA according to the manual instructions. Hybridization was carried out at 42 °C and washes were performed at 65 °C in 2 × SSC and 0.1 × SSC. The final immunological detection was performed following the manual instructions (Roche, DIG Northern Starter Kit). Images were captured with an ImageQuant LAS4000 Biomolecular Imager (GE Healthcare) (Supplementary Fig. 11).

**Ribosomal profile analysis**. The ribosomal profile was analyzed as following procedure[69]. One-day-old adult worms (100 μl) were collected on ice and washed several times with M9 buffer, and then homogenized in 700 μl lysis buffer (10 mM Tris-Cl pH 7.4, 5 mM MgCl$_2$, 100 mM KCl, 1% (v/v) Triton X-100, 0.5% (w/v) deoxycholate, 1 U ml$^{-1}$ RNase inhibitor, 2 mM dithiothreitol, and 0.1 mg ml$^{-1}$ cycloheximide) on ice. After complete homogenization, the worm samples were briefly vortexed, incubated on ice for 10 min, and then centrifuged at 4000 r.p.m. for 10 min at 4 °C. The supernatant was transferred to a new tube. Sucrose gradients of 10–50% were generated at 4 °C with a Gradient Master (BioComp Instruments). Next, 600 μl supernatant (homogenized sample, normalized to the absorbance at 260 nm using cytoplasmic RNA) was applied to the top of the

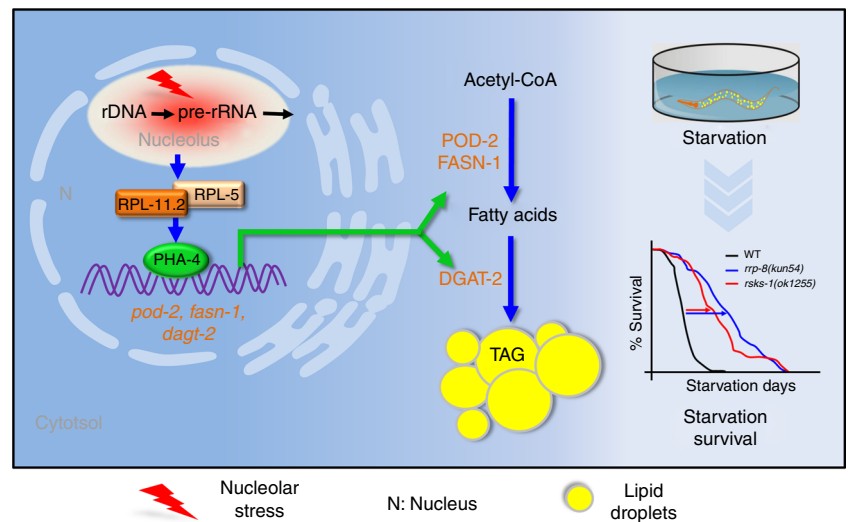

**Fig. 8** A proposed model of the RPL-11.2/RPL-5-PHA-4-mediated lipogenesis pathway required for nucleolar stress-induced lipid accumulation. Nucleolar stress is triggered by perturbation of pre-rRNA transcription and processing, activates PHA-4 expression, which is mediated by RPL-11.2/RPL-5 in *C. elegans*. Both RPL11 and RPL5 are conserved ribosomal proteins that are required for the nucleolar stress response in mammalian cells. PHA-4 subsequently binds to and transactivates the expression of the lipogenic genes *pod-2* encoding acetyl-CoA carboxylase, *fasn-1* encoding fatty acid synthase, and *dgat-2* encoding diacylglycerol *O*-acyltransferase 2. Upregulated POD-2, FASN-2, and DGAT-2 expression further promotes the biosynthesis of fatty acids and triacylglycerol (TAG), leading to excessive lipid accumulation. Furthermore, the increased lipid accumulation promotes worm survival under starvation

**Fig. 7** Nucleolar stress induces *dgat-2* expression to promote lipid accumulation. **a** Confocal microscopy of DGAT-2::GFP in WT, *rrp-8*(*kun54*), *rsks-1* (*ok1255*), and AD-treated worms. Scale bar represents 20 μm in the panel. **b** Relative fluorescence intensity of DGAT-2::GFP quantified from **a**. Data are presented as the means ± SD of at least 20 worms for each worm strain. Significant difference between WT and a specific worm strain, Student's *t*-test, ***$P < 0.001$. **c** Immunoblotting of DGAT-2::GFP. One-day-old WT, *rrp-8*(*kun54*), *rsks-1*(*ok1255*), and AD-treated worms expressing DGAT-2::GFP were collected for lysing and immunoblotting with anti-GFP antibody. **d** Confocal microscopy of DGAT-2::GFP fluorescence in WT, *rrp-8*(*kun54*), *rsks-1*(*ok1255*), and AD-treated worms treated with either empty vector (EV) or *pha-4* RNAi from L1 to day 1 of adulthood. Scale bar represents 20 μm. **e** Relative fluorescence intensity of DGAT-2::GFP quantified from **d**. Data are presented as the means ± SD of at least 20 worms for each worm strain. Significant difference between WT and a specific worm strain, Student's *t*-test, ***$P < 0.001$. Significant difference between a specific strain treated with empty vector (EV) and *pha-4* RNAi, ###$P < 0.001$. **f** Schematic diagram of the generation of *kun140* and *kun141* mutations of *dgat-2* using multi-sgRNA-directed CRSPR/cas-9 technology (upper panel). Deletion of the *kun140* and *kun141* mutations of *dgat-2* was confirmed using PCR (lower panel). **g** Nile Red staining of fixed 1-day-old adult worms. Representative animals; the anterior is indicated on the left and the posterior is indicated on the right. Scale bar represents 20 μm. **h** Distribution of the lipid droplet size (% lipid droplets) measured by Nile Red staining of fixed worms from **g**. Data are presented as the means ± SD of 10 animals for each worm strain. Significant difference between a specific mutant strain without (EV) and with RNAi treatment, Student's *t*-test, ***$P < 0.001$, **$P < 0.01$, *$P < 0.05$. **i** Percentage of triacylglycerol (% TAG) in total lipids (TAG + phospholipids, PL) analyzed by TLC/GC. Data are presented as the means ± SD of four biological repeats. Significant difference between a specific mutant strain with and without *dgat-2* mutant background, Student's *t*-test, **$P < 0.01$, *$P < 0.05$

sucrose gradients for ultracentrifugation (Beckman SW40 rotor) at 36,000 r.p.m. for 3 h at 4 °C. Gradients were fractionated with continuous monitoring from bottom to top based on the absorbance at 254 nm.

**ChIP assay**. ChIP assay was performed as previously described[70] with minor modifications. Young adult worms were collected and frozen in an equal volume of M9 buffer at − 80 °C and subsequently ground into a fine frozen powder. After crosslinking, the pellets were washed with cold M9 buffer and suspended in FA buffer for sonication using a bioruptor at 4 °C for 20 cycles (30 s on after 30 s off). Lysates were immunoprecipitated with 5 μg of ChIP-grade anti-GFP antibody (ab290, Abcam) and IgG (ab172730, Abcam). Subsequently, 40 μl of protein A beads (ab193255, Abcam) was added to each ChIP sample, followed by washing with FA buffer, FA-500 mM buffer, LiCl buffer, and TE buffer. The immuno-complexes were eluted with fresh ChIP elution buffer (1% SDS and 0.1 M NaHCO₃), and the crosslinking was reversed. DNA fragments were purified using the Omega DNA purification column (D6492, omega) and subsequently used for QPCR analysis.

**Visualization of GFP fluorescence**. At least 20 GFP worms were picked, mounted on an agarose pad, and anesthetized with 10 mM sodium azide. GFP fluorescence was visualized under an OLYMPUS BX53 fluorescence microscope (Olympus, Japan) and quantified in the anterior of the intestine using a fixed exposure time.

**Western blot analysis**. One-day-old adult worms were collected in M9 buffer and quickly washed several times on ice. To each worm sample was added moderate high-salt RIPA, followed by homogenization at 4 °C. The tissue homogenates were then centrifuged and the supernatants were used for western blot analysis. The total proteins were denatured for 5 min at 95 °C. Equal amounts of protein samples were loaded and separated by 12% SDS-PAGE gel (Bio-Rad) and then transferred to polyvinylidene difluoride membranes. The primary antibodies were rabbit anti-GFP antibody (Abcam) and anti-Actin-β (Sigma) at 1:1000 dilutions. The secondary antibody was goat anti-rabbit and goat anti-mouse IgG from Beyotime at 1:5000 dilutions. Images were captured with an ImageQuant LAS4000 Biomolecular Imager (GE Healthcare) (Supplementary Fig. 12).

**Worm survival under starvation**. Approximately 120 synchronized L4 worms feeding on NGM plates were picked, transferred to M9 buffer without food, and cultured at 20 °C. The number of live worms was counted every day.

**Statistical analysis**. Data are presented as the means ± SD unless specifically indicated. Statistical analyses included *t*-tests or analysis of variance. All figures were generated using GraphPad Prism 6 (GraphPad Software, La Jolla, CA, USA) and Photoshop CS4.

**Data availability**. The authors declare that all data supporting the findings of this study are available within the article and its Supplementary Information files. The RNA-seq data are deposited in figshare with the identifier [https://doi.org/10.6084/m9.figshare.5808243] or [data source: https://figshare.com/s/87b306c73a81b20e9624].

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

## Acknowledgements

We thank Dr Souhong Guang for kindly providing worm strain EG4322, plasmid pCFJ151, pJL43.1, and CRISPR/cas-9 technical support, and Dr Ge Shan for support in northern blotting. We thank Xu Liu for northern blot technical assistance, and Kang Xie and Dejiu Zhang for ribosomal profile analysis. Some strains were provided by the CGC, which is funded by NIH Office of Research Infrastructure Programs (P40 OD010440). This work was supported by the Strategic Priority Research Program of the Chinese Academy of Sciences (XDB13030600), National Natural Science Foundation of China (U1702288, 31671230, 31160216, 31600963, 31460268, U1402225), Yunnan Natural Science Foundation (2017FA007), and Yunnan Provincial Science and Technology Department (2014HB022), Yunnan Oversea High-level Talents Program (2015HA040).

## Author contributions

B.L. supervised the research. J.W. and B.L. conceived and designed the experiments. J.W. carried out most experiments and data analysis. X.J. analyzed lipid contents and assisted with strain construct and images processing. X.Z., J.W., X.J., Y.Z., Y.L., T.Z., Z.Z., L.Z., J.Z., and Y.W. contributed reagents/materials/analysis tools. J.W. and B.L. wrote the manuscript.

## Additional information

**Competing interests:** The authors declare no competing interests.

