## [Peer Review File(PDF 2577 kb) · Nature Communications]

Reviewer #1 (Remarks to the Author):

Wu and colleagues present a potentially interesting story suggesting a pathway for intestinal lipid modulation in response to nucleolar stress. Nucleolar stress can be induced by down regulation of rRNA processing or transcription, and this leads to an increase in the size of lipid droplets. The authors argue that alterations in lipid droplets requires direct regulation of the transcription factor PHA-4.

- A main question is what biological purpose is served by changes in lipid metabolism in response to changes in ribosomal biogenesis. Does this help worms survive nucleolar stress?
- A second concern is the nature of the lipid changes. The most dramatic change is the increase in droplet size from 1um to 3 um. Most of the other changes are extremely small e.g. the triglyceride measurements. Part of the problem may be the reliance on Nile Red for many experiments. Nile Red stains lysosome-related organelles and misses most lipids (e.g. Schroeder et al., MBoC 2006 and O'Rourke et al., 2009). The authors should use Oil Red O and/or lipid fractionation for all assays. The authors should show a lipid profile, and it would be very interesting to see if the types of lipids changed. This might give the authors a better assay for all their experiments.
- A third question relates to PHA-4 regulation, and the issue of parallel vs linear pathways. The authors convincingly show that PHA-4::GFP can bind to the tested regions, but two additional experiments are needed to conclude that these genes are modulated by PHA-4 in response to stress. First, the authors need to include some other promoters (negative and positive controls) that do not change with stress and are not involved in lipid accumulation. Second, because PHA-4 binds thousands of places in the genome, the authors need to mutate one or more PHA-4 binding sites and show that the binding matters for induction of the gene of interest. This is important because the intestines look small in the images, which could derive from poor feeding due to pha-4 inactivation (i.e. an indirect effect). The authors need to show the effect is direct.

Technical aspects:

- Alternative measures of lipids for all figures
- Controls for CHIP
- Inactivation of PHA-4 binding sites to test for direct induction
- The methods and legends are extremely brief and it is often difficult to tell what the authors did. Please expand the methods in the Supplemental Data. In particular, how were results normalized? It is often impossible to tell if the quantitation is reliable (levels of PHA-4 etc.)

- Note that Figure 1-I looks like a nucleolar localization pattern to my eye, not just nuclear. This should be confirmed with a nucleolar co-stain.
- Please show standard deviation as a better measure of variability than SEM.

Reviewer #2 (Remarks to the Author):

Wu et al reported the role of PHA-4, a FOXA class transcription factor, in linking nucleolar stress to a fat anabolic pathway in *C. elegans*. This study stemmed from the observation that a defect in ribosomal RNA processing caused lipid accumulation. In mammals, nutrient dependent regulation of ribosome assembly is dependent on p53. However, after surveying a number of transcription factors, including the *C. elegans* orthologs of p53, SREBP and FOXO, the authors concluded that PHA-4/FOXA was responsible for the activation of a number of genes that increased triglyceride synthesis, downstream of nucleolar stress. The genetic analysis in this manuscript was comprehensive. However, the authors failed to mention that *pha-4* mutant worms are sick due to developmental defects and it was not clear how such 'general sickness' contributed to reduced fat synthesis. Although standard methods for the quantitative analysis of lipids in *C. elegans* were employed, the authors should provide better description of these methods to allow general readers to judge how results from *C. elegans* can be compared with previous studies in mammalian systems. A more comprehensive discussion on the use of distinct transcriptional programs downstream of nucleolar stress in *C. elegans* and mammals should also be included. Would FOXA be engaged in mammals under specific regime of nucleolar stress? Is there a difference between proliferative and differentiated cells?

Major points:

(1) The triglyceride (TAG) measurements were all reported as % total lipids. Although this method was used in previous publications, it did not report if the absolute amount of TAG has changed. Instead, the numbers seemed to reflect changes in the relative abundance of TAG versus phospholipids. At the very least, the authors should spell out their method of analysis and state the potential complication in data analysis, e.g. when there is a simultaneous increase in both TAG and phospholipids.

(2) Would alternative triggers of nucleolar stress (other than *rrp-8*, *pro-2/3*, *rsk-1* mutations, actinomycin D treatment) engage the *C. elegans* p53 ortholog? Readers who are familiar with the literature on p53 and nucleolar stress in mammals may find it difficult to reconcile published results with those in this manuscript.

(3) It was unclear if GFP fusion proteins of RRP-8, PHA-4 and DGAT-2 were fully functional. Could they rescue *rrp-8*, *pha-4* and *dgat-2* mutants? Since the authors used these fusion proteins to study their localization and regulation, it is paramount to know that they can fully substitute the endogenous proteins.

(4) Figure 4G is confusing. Which ones were specific bands? It was hard to judge without a negative control. The relative abundance of different PHA-4 isoforms was also inconsistent in actinomycin D treated, and *rrp-8* and *rsk-1* mutant worms. Did it reflect differential regulation of PHA-4 isoforms?

(5) Figure 5C and 5D lacked clarity. It was not obvious that “Pdgat-2 #2” meant primer pair #2 was used for ChIP at the *dgat-2* promoter. Such details were missing in the figure legend. The primer sequences and product size for each primer pair used should be included in the manuscript.

(6) The images shown for DGAT-2::GFP in Figure 6A, 6D and S7 did not reflect the correct localization of DGAT-2. The signals appeared diffuse in the cytoplasm, which might affect the accuracy of fluorescence intensity measurement. As a result, the large increase in DGAT-2::GFP level as indicated by Western blotting (Figure 6C) did not correlate well with data in Figure 6B.

Minor points:

(1) p.11 “the mRNA level of *pha-4* (Figure 4D)” should be “the mRNA level of *pha-4* (Figure 4E)”.

(2) p.11 “the fluorescence of PHA-4::GFP (Figure 4E)” should be “the fluorescence of PHA-4::GFP (Figure 4F)”.

(3) Supplemental information, line 21. The convention for naming single copy transgenes should be ‘Si’ instead of ‘Is’. Therefore, *kunIs124* should be *kunSi124*.

Reviewer #3 (Remarks to the Author):

In this work, Wu et al described a pathway that linked nucleolar stress with lipid accumulation in *C.elegans*. It is shown that alteration in ribosome biogenesis (rRNA transcription and processing) leads to increased lipid accumulation and triacylglycerol (TAG) content. They identified the transcription factor PHA-4 that acts as a sensor of nucleolar stress to bind and transactivate the expression of the lipogenic genes *pod-2*, *fasn-1* and *dgat-2*, with consequent increase in lipid biosynthesis and accumulation.

This study initiates with the observation that a mutant for *rrp8* displayed enlarged lipid droplets and increased lipid accumulation and triacylglycerol (TAG) content. In yeast RRP-8 is a nucleolar protein with methyltransferase activity for m1A base modification of 25S rRNA, which was implicated in pre-rRNA cleavage. Its mammalian homolog NML was also reported to modify rRNA. Based on this information the authors described that *rrp8* mutants showed defects in rRNA processing and display a polysome profiles with a decrease in 60S subunits and polysome levels. Interestingly, treatment with Actinomycin D (Act D), an inhibitor of rRNA gene transcription also induced lipid accumulation but did not further enhance lipid accumulation in *rrp8(kun54)*, suggesting that RRP-8 and Act D act in one pathway, that is ribosome biogenesis. Similarly to *rrp8* mutants, also mutants for *rsks-1*, a homolog of the p70 ribosomal protein S6 kinase and effector of the TOR pathway, showed decreased polysome levels and 60S subunits but also increased lipid accumulation. Because of these results, the authors defined these conditions (*rrp8* and *rsks-1* mutants and ActD treatment) as nucleolar stress.

The authors then analyzed three factors EGL-9, AAK-2, PHA-4, which were described to mediate lifespan extension of the *rsks-1(ok1255)* mutant. They found that only PHA-4 is required for lipid accumulation observed in *rrp8* and *rsks-1* mutants and ActD-treated worms. Moreover, PHA-4 expression was increased in *rrp8* and *rsks-1* mutants and ActD-treated worms. Gene expression analysis revealed that transcription of lipid metabolic genes *fasn-1*, *pod-2* and *dgat-2* was upregulated in *rrp8* and *rsks-1* mutants and ActD-treated worms whereas their expression is suppressed in *pha-4* mutants. The data revealed that PHA-4 associates with the promoter region of *fasn-1*, *pod-2* and *dgat-2* and this association increase in *rrp8* and *rsks-1* mutants. RNAi kD of *fasn-1* and *pod-2* in *rrp8* and *rsks-1* mutants at L2/L3 developmental stage resulted in low lipid accumulation. Thus, the activities of FASN-1 and POD-2 are required for the increase in lipid droplets in *rrp8* and *rsks-1* mutants. The authors then analyzed DGAT-2, which is described to be required for lipid droplet expansion, and found that it is higher expressed in *rrp8* and *rsks-1* mutants and ActD - treated worms and downregulated upon RNAi knockdown of *pha-4*. Finally, KO of DGAT-2 suppressed lipid accumulation in *rrp8* and *rsks-1* mutants.

The accumulation of liquid droplets upon impairing rRNA transcription by ActD treatment and in *rrp-8* and *rsks-1* mutants is clear, interesting and novel. However, the work as it is remains descriptive and the mechanism by which ribosome biogenesis affects lipogenesis pathway are elusive. The data point to a role of PHA-4 as a nucleolar stress sensor that is required for nucleolar stress-induced lipid accumulation. However, how PHA-4 is regulated by defects of ribosome biogenesis is unclear and not addressed. Moreover, the hypothesis is that under nucleolar stress excessive energy is stored as fat. Why? Which are the consequences when this storage is inhibited? Finally, the description of how the experiments have been performed is very often lacking important details, making difficult to evaluate the results.

Fig. 1

I liked very much how the mutant *kun54* has been analyzed.

In Fig. 1 it is described that RRP8 is ubiquitously expressed in almost all cells throughout all developmental stages. I see more, it appears to me that RRP8 is in the nucleolus. However, this specific localization is lost in Figure S1 due to high exposure. The experiment of Figure S1 is important since it serves to determine that the expression level or nuclear localization of WT and the mutant RRP-8 (G301R)::GFP is the same. A less exposure of this image might have revealed differences in nucleolus/cellular localization between WT and mutant RRP-8 such as a delocalization from the nucleolus.

In Fig. 2 it is described that RRP8 affects rRNA methylation and consequently rRNA processing. The qPCR is a nice method but for analysis of defects in rRNA processing a Northern analysis is an obligatory step. The same is true for the measurements of pre-rRNA. In Fig. 2G the values are detected by qPCR. How have been these data calculated? Short legends such as "Relative rRNA levels detected by qPCR." are not of help.

In Fig. 3 polysome profile data are shown. The decrease in 60S subunit in *rrp-8* and *rsks-1* is for me undetectable. What is relevant here is the decrease in polysomes. Because of defects in rRNA processing described in Fig. 2, one should expect a decrease in 80S levels, which is however not the case. Why? Moreover, the description of ribosome profile in M&M does not contain the necessary information to evaluate the data. How have the samples been loaded on the gradient? Same amount of cells, proteins or cytoplasmic RNA? How the peaks have been measured? This is particularly relevant for the very little peak of 60S. I have also some concerns about the treatment with Act D. How long was the treatment? Act D inhibits transcription since is an intercalator with preference for CG-rich sequences. Correct dosage of Act D is key to study rDNA transcription. 50 ng/ml is the concentration of ActD used to analyze rRNA gene transcription since rRNA genes in mammals have a high content of CG. The concentration here used was 15 ng/ μ L. Why this concentration? And how long have been treated the worms? This is an important point since elevated ActD concentration inhibits transcription from both Pol II and Pol I genes. If this is the case the ActD treatment is not anymore specific for rRNA genes.

The statement "suggesting that Actinomycin D plays an evolutionarily conserved function to inhibit rDNA transcription from worms to mammals" should be toned down. First, no data on rRNA gene transcription are shown. Second, a chemical compound cannot play an evolutionary conserved function.

In Fig. 4 it is described the identification of PHA-4. The data clearly showed that the formation of lipid droplets in *rrp-8* and *rsks-1* mutants and ActD treated worms require the expression of PHA-4. However, there is a lack of information how the experiments have been performed. It is described that "The temperature-sensitive strain *pha-4(zu225);smg-1(cc546ts)* showed inactivation of PHA-4 under a restrictive temperature of 20°C". No data have been shown on this or reported through reference. Data of Fig. 4C showing % TAG in lipids in the *smg-1(cc546ts)* mutation alone should be integrated in Fig. 4D or at least use the same Y axis scale. At which temperature the analysis of the *smg-1(cc546ts)* control has been performed?

The increase expression of PHA-4 mRNA in *rrp-8* and *rsks-1* mutants and ActD treated worms is very clear. However, I do not see any dramatic increase in "fluorescence of PHA-4::GFP (*unc-119(ed3);wglis37*) (Fig.4E)". In contrast, it appears that cells are enlarged. The western in Fig. 4F lacks clarity and the short legend "Immunoblotting of PHA-4::GFP with anti-GFP antibody." does not help. What are a, b and c? How the valued of fold expression have been calculated? Why are there so many bands? I will also be careful in the normalization considering that the defects observed in polysome profiles might affect the total amounts of proteins (see the 4.4 fold change in the ActD sample). Finally, it is indicated that "PHA-4::GFP is expressed in the nucleus of intestine cells, similar to nuclear expression of RRP-8::GFP. What is relevant here? The nuclear expression or the expression in intestine cells? Please clarify.

What is unclear here is how PHA-4 get activated, an issue that is ignored in the whole manuscript.

In Fig. 5, the analysis of RNAseq in *rrp8* mutants revealed the activation of *dgat-2*, *fasn-1* and *pod-2* genes, which are implicated in lipogenesis. Expression of these genes depends on PHA-4 levels. The PHA-4 ChIP analysis indicated PHA4 bind the promoter of these genes. I would like to see here further controls to substantiate the specificity of this assay such as amplifications of genes with transcription not affected by *rrp8*, *rsks-1* and AD treatment (qRT-PCR, Fig. 5B) and sequences not bound by PHA-4 (ChIP, Fig. 5E-F).

In Fig. 6 it is described DGAT-2. The data in Fig. 6A are very clear and support the data of Fig. 5 showing that DGA2 expression depends of PHA-4 expression. It is however not a surprise that deletion of DGAT-2 impairs accumulation of lipid droplets in in *rrp-8* and *rsks-1* mutants and ActD treated worms. Indeed, as indicated by the authors DGAT-2 catalyzes the conjugation of a fatty acyl-CoA to diacylglycerol (DAG) to form TAG and is a lipid droplet (LD) protein required for LD expansion. It might have been interesting here to determine what happens to the lipid droplets through KD of another gene implicated in lipogenesis but with expression not altered upon ribosome biogenesis

(i.e. *rrp8* mutants). Finally, which are the consequences to abolish lipid droplets upon "nucleolar stress"?

Minor points

The analyses of *cep1* mutants in lipid formation are in Fig. S2C and S2D (and not S3C and S4D).

Fig 2C represents 6n values which are not shown in Fig. 2B. 80S values should be shown.

Response to reviewers

Reviewer #1 (Remarks to the Author):

Wu and colleagues present a potentially interesting story suggesting a pathway for intestinal lipid modulation in response to nucleolar stress. Nucleolar stress can be induced by down regulation of rRNA processing or transcription, and this leads to an increase in the size of lipid droplets. The authors argue that alterations in lipid droplets requires direct regulation of the transcription factor PHA-4.

We very much appreciate your comments on that we present a potentially interesting story. We will address your suggestions or comments one by one below.

- A main question is what biological purpose is served by changes in lipid metabolism in response to changes in ribosomal biogenesis. Does this help worms survive nucleolar stress?

Answer:

Nice suggestion. The answer is Yes.

Many studies have shown that animals/cells often shift their metabolism to increased nutrient storage and efficiency of energy utilization under harsh conditions like prolonged starvation. The nucleolus is a central hub for coordinating the stress response, since nearly all metabolic and signaling pathways ultimately lead to or from it.

Our present study found that nucleolar stress promotes lipid accumulation; we consequently thought whether nucleolar stress can help worm to live longer under starvation, in order to address this question. We then performed starvation experiments by leaving synchronized L4 worms in M9 buffer without food, and counting living animals every day. Indeed, we found that *rrp-8(kun54)* and *rsk-1(ok1255)* mutant worms, as well as AD treated worms, displayed extended survival compared with wild type (WT) worms under starvation condition, suggesting that nucleolar stress is beneficial for worms survival. Meanwhile, we found that the increased survival depends on excessive lipid accumulation because mutation of the *dgat-2* abolished this benefit. We added these results as Figure S10 in the revised manuscript.

During the revision, we found a report by Dr. Alexander Soukas's group. They also showed that reduced function of cytoplasmic aminoacyl tRNA synthetases (ARS genes) leads to increased fat storage and extended starvation survival in *C. elegans*. Consistently, a very early report also showed that genetic activation of protein synthesis results in opposite effects in *Drosophila*. Therefore, metabolic alteration via reduced ribosome biogenesis or protein production may indeed help animals to survive under starvation condition.

- A second concern is the nature of the lipid changes. The most dramatic change is the increase in droplet size from 1um to 3 um. Most of the other changes are extremely small e.g. the triglyceride measurements. Part of the problem may be the reliance on Nile Red for many

experiments. Nile Red stains lysosome-related organelles and misses most lipids (e.g. Schroeder et al., MBoC 2006 and O'Rourke et al., 2009). The authors should use Oil Red O and/or lipid fractionation for all assays. The authors should show a lipid profile, and it would be very interesting to see if the types of lipids changed. This might give the authors a better assay for all their experiments.

Answer:

It is now widely known that the staining of living *C. elegans* with several dyes like Nile Red, Bodipy, does not stain lipid droplets. On the contrary, staining of fixed worms with these dyes do stain lipid droplets, which have been confirmed and being used by several labs including us since 2009. When I was a postdoctoral fellow in Dr. Jennifer Watts' lab, almost at the same time as O'Rourke et al. did, we showed that Nile Red staining of living worms did not correlate to the chromatography methods (Brooks et al., 2009; Liang et al., 2010). In these two papers, we further developed a new method to fix worms first and then to stain lipid droplets with Nile Red dye. This Nile Red staining of fixation has been further confirmed to indeed stain lipid droplets by several labs, e.g. Dr. Ho Yi Mak' lab (Zhang et al., 2010), Dr. Heidi Tissenbaum' lab (Yen et al., 2010), Dr. Pingsheng Liu's lab (Na et al., 2015; Zhang et al., 2012). Recently, Dr. Pingsheng Liu's lab showed that Nile Red, LipidTox, Bodipy, and Oil Red, all four dyes post-fixation gave signals that were well co-localized with lipid droplet protein maker DHS-3 (Na et al., 2015).

In this manuscript, like others papers we had published, we used Nile Red of fixation, not the Nile Red of living, to show lipid droplets and storage in *C. elegans*. We are pretty cautious about the methodologies in our research. In my lab, we must use at least two different methods to double detect lipid droplets and fat storage, often one is a lipid dye, the other is the chromatography method. In the past several years, my lab has tested around 100 *C. elegans* mutants or RNAi worms, including several well-known high fat or low fat mutants such as *daf-2*, *rsk-1*, *daf-7*, *pept-2*, *fat-6;fat-7*, *sbp-1*, and new mutations from our whole genome screen by EMS (our unpublished data). Both the Nile Red staining of fixation and the chromatography method have displayed pretty consistent and reproducible results in all these mutants or RNAi worms.

When we isolated *kun54* mutant, we used Nile Red staining (Figure 1A), LipidTox Red staining (Figure 1B), and Oil red staining (Figure 1C) of fixation, as well as TLC/GC chromatography method (Figure 1E), to investigate its lipid phenotypes. All four methods consistently showed that *kun54* mutant has increased lipid accumulation compared with WT. In the following experiments, we used both Nile Red staining of fixation and TLC/GC chromatography method to demonstrate lipid phenotypes. TLC/GC chromatography method is a very time-consuming and heavy work that needs more than 40,000 worms for each biological repeat.

Some *C. elegans* laboratories used total proteins or dry weight to normalize TAG content, which often give a big change of TAG in worms. We have tested these methods, but found that they are often variable because the analysis approaches of lipids and total proteins or dry weight are separated and independent. Instead, %TAG [TAG/(TAG+PL)] is much highly stable and reproducible in many worm strains we have testes so far over ten years, since we extract, separate, and analyze TAG and PL at same time and same condition.

As you can see from below TLC plate (**Response Figure 1**) that is used to separate different lipids, the levels of PL (phospholipids) are relatively stable in WT and *pha-4(zu225)* and *rrp-8(kun54)* worms, but the amount of TAG is obviously different among these worm strains. Generally, the amount of TAG+PL is >90% in total lipids in worms, and the amount of TAG is about half (48-52%) in TAG+PL in wild type (WT) worms, based on our numerous measurements. Since TAG level is also used in denominator [TAG/(TAG+PL)] to calculate TAG content, the change of %TAG looks somewhat small consequently. However, for example, 5% increase of TAG in total TAG+PL means more than 10% TAG content increase at least, which may be a lot for a tiny worm.

Response Figure 1. Lipid separation by thin-layer chromatography in WT, *pha-4(zu225)* and *rrp-8(kun54)* worms. TAG: triacylglycerol; Ch: cholesterol; DAG: diacylglycerol; PL: phospholipids. Each lane represents a biological repeat.

- A third question relates to PHA-4 regulation, and the issue of parallel vs linear pathways. The authors convincingly show that PHA-4::GFP can bind to the tested regions, but two additional experiments are needed to conclude that these genes are modulated by PHA-4 in response to stress. First, the authors need to include some other promoters (negative and positive controls) that do not change with stress and are not involved in lipid accumulation. Second, because PHA-4 binds thousands of places in the genome, the authors need to mutate one or more PHA-4 binding sites and show that the binding matters for induction of the gene of interest. This is important because the intestines look small in the images, which could derive from poor feeding due to *pha-4*

inactivation (i.e. an indirect effect). The authors need to show the effect is direct.

Answer:

Following your suggestions, we performed below experiments to address these issues.

1) Negative and positive control.

Following your nice suggestion, we tested the promoters of two genes *taf-1* and *myo-2* as the negative and positive control of ChIP-QPCR, respectively (Figure S7A). *taf-1* encoding an ortholog of human TATA-binding protein associated factor TAF1L (TAFII250) and *myo-2*, encoding a muscle-type specific myosin heavy chain isoform were previously reported as the negative and positive control of PHA-4::GFP ChIP, respectively (Hsu et al., 2015). In addition, we also found that the expression of *taf-1* was unchanged in our transcriptome of wild type and *rrp-8(kun54)*. We added these data as Figure S7A in the revised manuscript.

2) PHA-4 binding sites.

Our ChIP-qPCR analyses showed the PHA-4 has ability to bind to the promoter regions of *fasn-1*, *pod-2* and *dgat-2* genes (Figure S7A), and the binding of PHA-4 to these genes showed an increased tendency in both the *rrp-8(kun54)* and *rsk-1(ok1255)* mutants compared with WT worms (Figure 6D-F). To test the binding for induction of these target genes, we opted to investigate the expression of *dgat-2*, since it involves in the last step of triacylglycerol (TAG) biosynthesis and its promoter has the minimum number of putative binding sites for PHA-4 compared with *fasn-1* and *pod-2* (Figure 6C). We generated four constructs with full-length promoter (*full*) or truncated promoters (*m1*, *m2*, *m1+m2*) of *dgat-2* fused with green fluorescent protein (GFP) as a reporter (Figure S7B). We found that the *m2*, but not the *m1* binding site of *dgat-2* promoter is crucial for its induction. These experiments including ChIP-qPCR analysis consistently show that *dgat-2* is a direct target of PHA-4, and PHA-4 can directly binds to the *m2* element to transactivate its expression under nucleolar stress. These data were added to revised manuscript as Figure S7B, D.

3) Poor feeding of *pha-4* mutant.

Dr. Susan Mango's laboratory has done a lot excellent works on PHA-4. They showed that PHA-4 plays critical role for pharyngeal development (Mango et al., 1994), raising question that the suppression of nucleolar stress-induced lipid accumulation in the *pha-4(zu225)* mutant might be due to its pharyngeal pumping defect. To test this possibility, we examined *eat-2*, which encodes a nicotinic acetylcholine receptor subunit that functions in the pharyngeal muscle, and mutation of *eat-2* exhibits feeding defect with reduced pharyngeal pumping (Avery, 1993; Lakowski and Hekimi, 1998; Panowski et al., 2007). Although *eat-2(ad465)* mutant worms showed decreased lipid accumulation compared with wild type (WT) worms, their TAG content and lipid droplet size were still increased in *rrp-8(kun54)* mutant background (Figure S6A-C). Therefore, we think that the suppression of nucleolar stress-induced lipid accumulation is probably not a result of feeding defect in *pha-4(zu225)* mutant. We added these data as Figure S6 in the revised manuscript.

Technical aspects:

- Alternative measures of lipids for all figures

Answer:

We measured the size of lipid droplets from the results of post-fixation of Nile Red staining and TAG content by TLC/GC for all mentioned worm strains in all figures.

- Controls for CHIP

Answer:

We used *taf-1* as the negative control and *myo-2* as the positive control, respectively, based on the reports by (Hsu et al., 2015). We added these results to Figure S7A in the revised manuscript.

- Inactivation of PHA-4 binding sites to test for direct induction

Answer:

We tested two PHA-4 binding sites of *dgat-2* (Figure S7B, C) and found that the *m2*, but not the *m1* binding site of *dgat-2* promoter is crucial for direct induction. We added these experiment information and results in the revised manuscript.

- The methods and legends are extremely brief and it is often difficult to tell what the authors did. Please expand the methods in the Supplemental Data. In particular, how were results normalized? It is often impossible to tell if the quantitation is reliable (levels of PHA-4 etc.)

Answer:

We added more detailed information to Methods and Materials as well as Legends, some are added to Supplemental information.

In addition, we added two Supplementary tables that list complete information of worm strains and primers.

- Note that Figure 1-I looks like a nucleolar localization pattern to my eye, not just nuclear. This should be confirmed with a nucleolar co-stain.

Answer:

Thanks for your sharp eyes. Yes. It is a nucleolar localization. Following your excellent suggestion, we confirmed that using mCherry::FIB-1 encoding the nucleolar protein fibrillarin to show that RRP-8 is expressed in the nucleolus in *C. elegans*. We added these data to Figure II in the revised manuscript.

- Please show standard deviation as a better measure of variability than SEM.

Answer:

We are very sorry for these mistakes. We actually used SD, but not SEM, in all data. We corrected all in the revised manuscript.

Reviewer #2 (Remarks to the Author):

Wu et al reported the role of PHA-4, a FOXA class transcription factor, in linking nucleolar stress to a fat anabolic pathway in *C. elegans*. This study stemmed from the observation that a defect in ribosomal RNA processing caused lipid accumulation. In mammals, nutrient dependent regulation of ribosome assembly is dependent on p53. However, after surveying a number of transcription factors, including the *C. elegans* orthologs of p53, SREBP and FOXO, the authors concluded that PHA-4/FOXA was responsible for the activation of a number of genes that increased triglyceride synthesis, downstream of nucleolar stress. The genetic analysis in this manuscript was comprehensive.

We very much appreciate your comments on our finding of PHA-4 linking nucleolar stress to a fat anabolic pathway. We will address your suggestions or comments one by one below.

However, the authors failed to mention that *pha-4* mutant worms are sick due to developmental defects and it was not clear how such ‘general sickness’ contributed to reduced fat synthesis.

Answer:

Yes. *pha-4* mutant worms are sick due to developmental defects.

Please also see our detailed response to Reviewer #1 regarding same concern on *pha-4* mutant.

Dr. Susan Mango’s laboratory has done a lot excellent works on PHA-4. They have shown that PHA-4 plays critical role in pharyngeal development (Mango et al., 1994), raising a question that the suppression of nucleolar stress-induced lipid accumulation in the *pha-4(zu225)* mutant might be due to its pharyngeal pumping defect. To address this issue, we examined *eat-2*, which encodes a nicotinic acetylcholine receptor subunit that functions in the pharyngeal muscle, and mutation of *eat-2* exhibits feeding defect with reduced pharyngeal pumping (Avery, 1993; Lakowski and Hekimi, 1998; Panowski et al., 2007). Although *eat-2(ad465)* mutant worms showed decreased lipid accumulation compared with wild type (WT) worms, their TAG content and lipid droplet size were still increased in *rrp-8(kun54)* mutant background (Figure S6A-C). Therefore, we think that the suppression of nucleolar stress-induced lipid accumulation is probably not a result of feeding defect in *pha-4(zu225)* mutant. We added these data as Figure S6 in the revised manuscript.

Although standard methods for the quantitative analysis of lipids in *C. elegans* were employed, the authors should provide better description of these methods to allow general readers to judge how results from *C. elegans* can be compared with previous studies in mammalian systems.

Answer:

We added more detailed information of lipid analysis to Materials and Methods titled “**Lipids extraction, separation and analysis**”, and also “**Vital dyes staining of lipid droplets (LDs)**” in Supplemental information.

A more comprehensive discussion on the use of distinct transcriptional programs downstream of nucleolar stress in *C. elegans* and mammals should also be included.

Answer:

We modified the model (Figure 8) and discussed these things in Discussion in the revised manuscript.

Would FOXA be engaged in mammals under specific regime of nucleolar stress? Is there a difference between proliferative and differentiated cells?

Answer:

There are three members of FOXA in mammals, FoxA1, FoxA2, and FoxA3, playing critical regulators of mammalian development and metabolism (Friedman and Kaestner, 2006). To examine whether they engage in nucleolar stress, we treated HepG2 cells with AD (50 ng/ml), a concentration often used in mammalian cells. We found that AD could increase lipid accumulation visualized by Oil Red O staining in HepG2 cells. And then, we checked the mRNA levels of these three genes, and found that the relative mRNA expression of both *FOXA1* and *FOXA3* was decreased by AD treatment. On the contrary, the relative mRNA expression of *FOXA2* was increased (below figure), which is like *pha-4* in *C. elegans*.

At this stage, we do not know much about whether FOXA engage in mammals under what specific regime of nucleolar stress, although the transcriptional expression of *FOXA2* is positively responsive to AD treatment; and is there a difference between proliferative and differentiated cells. These studies may be beyond the scope of this manuscript, although they are interesting questions for future study in mammalian systems. In this work, we are focusing on how PHA-4 responses to nucleolar stress to regulate lipid metabolism in model organism *C. elegans*.

Response Figure 2. AD treatment increased lipid accumulation and altered the expression of *FOXA1-3* in HepG2 cells. A: Oil Red O staining of HepG2 cells with or without AD treatment. B-D: Relative mRNA expression of *FOXA1*, *FOXA2* and *FOXA3*. Data are presented as the means \pm SD of three biological repeats. Significant difference between with and without AD treatment, *: $P < 0.05$, ***: $P < 0.001$.

Major points:

(1) The triglyceride (TAG) measurements were all reported as % total lipids. Although this method was used in previous publications, it did not report if the absolute amount of TAG has changed. Instead, the numbers seemed to reflect changes in the relative abundance of TAG versus phospholipids. At the very least, the authors should spell out their method of analysis and state the potential complication in data analysis, e.g. when there is a simultaneous increase in both TAG and phospholipids.

Answer:

It is now widely known that the staining of living *C. elegans* with several dyes like Nile Red, Bodipy, does not stain lipid droplets. On the contrary, staining of fixed worms with these dyes do stain lipid droplets, which have been confirmed and being used by several labs including us since 2009. When I was a postdoctoral fellow in Dr. Jennifer Watts' lab, almost at the same time as O'Rourke et al. did, we showed that Nile Red staining of living worms did not correlate to the chromatography methods (Brooks et al., 2009; Liang et al., 2010). In these two papers, we further developed a new method to fix worms first and then to stain lipid droplets with Nile Red dye. This

Nile Red staining of fixation has been further confirmed to indeed stain lipid droplets by several labs, e.g. Dr. Ho Yi Mak' lab (Zhang et al., 2010), Dr. Heidi Tissenbaum' lab (Yen et al., 2010), Dr. Pingsheng Liu's lab (Na et al., 2015; Zhang et al., 2012). Recently, Dr. Pingsheng Liu's lab showed that Nile Red, LipidTox, Bodipy, and Oil Red, all four dyes post-fixation gave signals that were well co-localized with lipid droplet protein maker DHS-3 (Na et al., 2015).

In this manuscript, like others papers we had published, we used Nile Red of fixation, not the Nile Red of living, to show lipid droplets and storage in *C. elegans*. We are pretty cautious about the methodologies in our research. In my lab, we must use at least two different methods to double detect lipid droplets and fat storage, often one is a lipid dye, the other is the chromatography method. In the past several years, my lab has tested around 100 *C. elegans* mutants or RNAi worms, including several well-known high fat or low fat mutants such as *daf-2*, *rsk-1*, *daf-7*, *pept-2*, *fat-6;fat-7*, *sbp-1*, and new mutations from our whole genome screen by EMS (our unpublished data). Both the Nile Red staining of fixation and the chromatography method have displayed pretty consistent and reproducible results in all these mutants or RNAi worms.

When we isolated *kun54* mutant, we used Nile Red staining (Figure 1A), LipidTox Red staining (Figure 1B), and Oil red staining (Figure 1C) of fixation, as well as TLC/GC chromatography method (Figure 1E), to investigate its lipid phenotypes. All four methods consistently showed that *kun54* mutant has increased lipid accumulation compared with WT. In the following experiments, we used both Nile Red staining of fixation and TLC/GC chromatography method to demonstrate lipid phenotypes. TLC/GC chromatography method is a very time-consuming and heavy work that needs more than 40,000 worms for each biological repeat.

Some *C. elegans* laboratories used total proteins or dry weight to normalize TAG content, which often give a big change of TAG in worms. We have tested these methods, but found that they are often variable because the analysis approaches of lipids and total proteins or dry weight are separated and independent. Instead, %TAG [TAG/(TAG+PL)] is much highly stable and reproducible in many worm strains we have testes so far over ten years, since we extract, separate, and analyze TAG and PL at same time and same condition.

As you can see from below TLC plate (**Response Figure 1**) that is used to separate different lipids, the levels of PL (phospholipids) are relatively stable in WT and *pha-4(zu225)* and *rrp(kun54)* worms, but the amount of TAG is obviously different among these worm strains. Generally, the amount of TAG+PL is >90% in total lipids in worms, and the amount of TAG is about half (48-52%) in TAG+PL in wild type (WT) worms, based on our numerous measurements. Since TAG level is also used in denominator [TAG/(TAG+PL)] to calculate TAG content, the change of %TAG looks somewhat small consequently. However, for example, 5% increase of TAG in total TAG+PL means more than 10% TAG content increase at least, which may be a lot for a tiny worm.

There may be a simultaneous increase in both TAG and phospholipids in an unknown specific worm strains. But in our study, the level of phospholipids is much stable in these strains (please see Response Figure 1).

We added more detailed information of lipid analysis in Materials and Methods titled “**Lipids extraction, separation and analysis**”.

(2) Would alternative triggers of nucleolar stress (other than *rrp-8*, *pro-2/3*, *rsks-1* mutations, actinomycin D treatment) engage the *C. elegans* p53 ortholog? Readers who are familiar with the literature on p53 and nucleolar stress in mammals may find it difficult to reconcile published results with those in this manuscript.

Answer:

We thank your critical comments. Initially, our works mainly focused on *rrp-8(kun54)*, *rsks-1(ok1255)* and AD treated worms, which showed increased CEP-1::GFP expression (Figure S3A and S3B). Then, we also wonder whether any other triggers of nucleolar stress also engaged on CEP-1. Subsequently, we detected CEP-1::GFP expression in *pro-2* and *pro-3* mutation worms, and found similar results in accordance with *rrp-8(kun54)*, *rsks-1(ok1255)* and AD treated worms (data not shown). Because there are many factors participating in the complicated processes of ribosomal biogenesis, we think other triggers of nucleolar stress may also engage the *C. elegans* p53 ortholog.

In *C. elegans*, *cep-1* is an ortholog of p53 in mammal. But CEP-1::GFP mostly express in germline of adult worms (20-40 hours post L4), which is different from p53 expression in almost all cells in mammals. As we know, *cep-1/p53* plays an important role in the regulation of apoptosis or proliferation. In *C. elegans*, germline is the primarily tissue accompany with apoptosis and proliferation in adult animals, which may be correlated with its expression pattern. Our current work focuses on the intestine, the major sites of fat synthesis and storage in *C. elegans*. These may hint why it is difficult to reconcile the results between p53 dependent nucleolar stress in mammal and *C. elegans*.

(3) It was unclear if GFP fusion proteins of RRP-8, PHA-4 and DGAT-2 were fully functional. Could they rescue *rrp-8*, *pha-4* and *dgat-2* mutants? Since the authors used these fusion proteins to study their localization and regulation, it is paramount to know that they can fully substitute the endogenous proteins.

Answer:

Overall, the answer is Yes. They could rescue *rrp-8*, *pha-4* and *dgat-2* mutants. We addressed these issues one by one.

1) Rescue of *rrp-8(kun54)* mutant by *rrp-8*.

In the very early stage when we cloned *kun54* as a mutation in *rrp-8* gene, we immediately did the rescue experiment. We injected wild type *rrp-8* gene mixed with fluorescence marker into the *rrp-8(kun54)* mutant worms. We observed three lines of transgenic strains and found that the increased lipid accumulation was obviously reversed in *rrp-8(kun54);kunEx193[Prnp-8::rrp-8]* mutant compared with *rrp-8(kun54)* mutant (**Response Figure 3**).

Response Figure 3. Nile Red staining of fixed worms. Represented animals; the anterior is indicated on the left and posterior is indicated on the right. Scale bar represents 20 μ m. #1-3 indicated three lines of transgenic worms of *rrp-8(kun54);kunEx193[Prp-8::rrp-8]*.

2) Rescue of *dgat-2(kun140)* mutant by *dgat-2::gfp*.

In this work, we generated two knocked-out alleles (*kun140* and *kun141*, Supplementary Table 1) of *dgat-2* using CRISPR/cas-9 technology (Figure 7F), and found that both the *kun140* and *kun141* mutations of *dgat-2* successfully suppressed increased lipid accumulation in *rrp-8(kun54)* and *rsk-1(ok1255)* mutant worms (Figure 7G-I). Meanwhile, we also generated an integrated translational strain of DGAT-2::GFP (*kunSi148[Pdgat-2::dgat-2::gfp]*) driven by its own promoter, and found that the fluorescence intensity of DGAT-2::GFP (Figure 7A and 7B) were dramatically increased in *rrp-8(kun54)*, *rsk-1(ok1255)*, and AD-treated worms.

To address above suggestion, we crossed *kunSi148[Pdgat-2::dgat-2::GFP]* into *rrp-8(kun54);dgat-2(kun140)* and *rsk-1(ok1255);dgat-2(kun140)* double mutants. We found that, compared with *dgat-2(kun140)* single mutant, *rrp-8(kun54);dgat-2(kun140);kunSi148* and *rsk-1(ok1255);dgat-2(kun140);kunSi148* mutants showed increased lipid accumulation and enlarged lipid droplets, which was similar to *rrp-8(kun54)* and *rsk-1(ok1255)* single mutant (**Response Figure 4**), suggesting that DGAT-2::GFP (*kunSi148[Pdgat-2::dgat-2::gfp]*) can rescue the response of *dgat-2* mutant to nucleolar stress.

Response Figure 4. Lipid accumulation of *dgat-2(kun140);kunSi148* in *rrp-8(kun54)* and *rsk-1(ok1255)* mutant backgrounds.

(A) Nile Red staining of fixed worms. Represented animals; the anterior is indicated on the left and posterior is indicated on the right. Scale bar represents 20 μm .

(B) Distribution of the lipid droplets size (% lipid droplets) as measured from Nile Red staining of fixed worms from (A). Data are showed as the means \pm SD of 10 animals for each worm strain.

3) Rescue of *pha-4(zu225)* mutant by *pha-4::gfp*.

According to previous report by Dr. Susan Mango's laboratory (Updike and Mango, 2007), *SM190[pha-4(zu225);smg-1(cc546ts)]* is a worm strain containing two genes mutation. Temperature sensitive mutant *smg-1(cc546ts)*, which is the component of nonsense-mediated decay (NMD) pathway, exhibits robust NMD activity at 15 $^{\circ}\text{C}$, but compromises activity at higher temperature. At restrictive temperature 15 $^{\circ}\text{C}$, high NMD activity of SMG-1(cc546) degrades *pha-4* mRNA from *pha-4(zu225)*, leading to lethality of *SM190* worms at L1 stage; while at 24 $^{\circ}\text{C}$, SMG-1(cc546ts) was compromised and *pha-4* mRNA is stabilized and accumulated, leading to survival of *SM190* worms. At 20 $^{\circ}\text{C}$, *SM190* presents an intermediate phenotype but lethal nonetheless.

To address whether *pha-4(zu225)* mutant could be rescued by *pha-4::gfp*, we generated *SM190;kunEx208[Ppha-4::pha-4::GFP,rol-6(su1006)]*. We transferred pregnant adults of *SM190* and *SM190;kunEx208[Ppha-4::pha-4::GFP,rol-6(su1006)]* to new NGM plates and raised at 20 $^{\circ}\text{C}$, and then observed their progenies 72 hours later. As showed in Response Figure 5, *SM190* worms displayed obvious arrest and even lethality (indicated by red arrow). Importantly, *SM190;kunEx208* worms could develop to adult worms. These results indicate that *pha-4::gfp* (*kunEx208*) can rescue the *pha-4(zu225)* mutation.

Response Figure 5. Development of different worm strains. Scale bar represents 500 μm .

(4) Figure 4G is confusing. Which ones were specific bands? It was hard to judge without a negative control. The relative abundance of different PHA-4 isoforms was also inconsistent in actinomycin D treated, and *rrp-8* and *rsk-1* mutant worms. Did it reflect differential regulation of PHA-4 isoforms?

Answer:

We thank for your critical comments and sorry for the confusion.

Response Figure 6. Genetic information of *pha-4* (www.wormbase.org)

Based on information provided by Wormbase (Response Figure 6), *pha-4* can be transcribed into three isoforms a (the longest one), b (the middle one), c (the shortest one). To clarify which isoform could respond to nucleolar stress, we generated three constructs *pha-4a::GFP*, *pha-4b::GFP* and *pha-4c::GFP* using individual transcripts of *pha-4* to fuse to GFP. We found that, similar to PHA-4::GFP (*wgIs37[pha-4::GFP]*) (Figure 4F), PHA-4a::GFP, PHA-4b::GFP and PHA-4c::GFP all express in nuclei (Response Figure 7). Based on western blot analysis, we could judge that PHA-4a::GFP is the primary band upregulated in *rrp-8(kun54)*, *rsks-1(ok1255)* and AD treated animals compared with wild type (WT) (Replicates 1, 2, 3, and 4 in Response Figure 8. Replicate 2 was shown in the initial submitted manuscript). The band sizes of PHA-4b::GFP and PHA-4c::GFP are very close, and also shorter than that of PHA-4a::GFP. It is pretty hard to detect PHA-4b::GFP and PHA-4c::GFP levels through several times of western blotting (Replicates 1 and 2 in Response Figure 8). Consistently, fluorescence microscopy also showed that PHA-4b::GFP and PHA-4c::GFP expression levels were obviously lower than PHA-4a::GFP (Response Figure 7). Therefore, we did not add PHA-4b::GFP and PHA-4c::GFP to Figure 4G in the revised manuscript.

In order to minimize the interference of unspecific bands, we optimized western blotting by prolonging blocking time (overnight) and also used new diluted anti-GFP antibody for immunoblotting (Replicate 3 and 4). In addition, we used WT worms without GFP as a negative control, and WT worms with PHA-4a::GFP $\{kunEx136[Pvha-6::pha-4a::GFP]\}$ as a positive control (Replicate 3 and 4, Response Figure 8). Consistently, the expression levels of PHA-4::GFP is obviously increased in *rrp-8(kun54)*, *rsks-1(ok1255)* and AD treated animals compared with WT animals. We replaced Figure 4G with the result of Replicate 4 in the revised manuscript.

Response Figure 7. The fluorescence of PHA-4a::GFP, PHA-4b::GFP and PHA-4c::GFP. *Pmyo-3::mCherry* was used as a transgenic marker and control.

Response Figure 8. Western blot of PHA-4a::GFP, PHA-4b::GFP and PHA-4c::GFP with anti-GFP antibody.

(5) Figure 5C and 5D lacked clarity. It was not obvious that “Pdgat-2 #2” meant primer pair #2 was used for ChIP at the *dgat-2* promoter. Such details were missing in the figure legend. The

primer sequences and product size for each primer pair used should be included in the manuscript.

Answer:

We included these information in the legends of Figure 6D in the revised manuscript. In addition, we listed the information of all primers for ChIP-QPCR used in this study in the Supplementary table 2.

(6) The images shown for DGAT-2::GFP in Figure 6A, 6D and S7 did not reflect the correct localization of DGAT-2. The signals appeared diffuse in the cytoplasm, which might affect the accuracy of fluorescence intensity measurement. As a result, the large increase in DGAT-2::GFP level as indicated by Western blotting (Figure 6C) did not correlate well with data in Figure 6B.

Answer:

The problem could be due to autofluorescence background in worm body. So, we redid these fluorescence experiments. We recaptured the DGAT-2::GFP using confocal microscope to maximally reduce the interference of autofluorescence in worm body and better reflect DGAT-2::GFP localization and intensity (Figure 7A, 7D and S9). Over again, we quantified DGAT-2::GFP fluorescence intensity from these confocal images and included the new results in the revised manuscript (Figure 7B and 7E). From our confocal images, as you can see that DGAT-2::GFP expression level drove by its own promoter was low in WT worms, and was significantly increased in *rrp-8(kun54)*, *rsk-1(ok1255)* and AD treated animals (Figure 7A). These new result may be well correlated with the western blotting result of DGAT-2::GFP with anti-GFP antibody (Figure 7C).

Minor points:

(1) p.11 “the mRNA level of pha-4 (Figure 4D)” should be “the mRNA level of pha-4 (Figure 4E)”.

Answer:

We corrected it in the revised manuscript.

(2) p.11 “the fluorescence of PHA-4::GFP (Figure 4E)” should be “the fluorescence of PHA-4::GFP (Figure 4F)”.

Answer:

We corrected it in the revised manuscript.

(3) Supplemental information, line 21. The convention for naming single copy transgenes should be ‘Si’ instead of ‘Is’. Therefore, kunIs124 should be kunSi124.

Answer:

We corrected it in the revised manuscript.

Reviewer #3 (Remarks to the Author):

In this work, Wu et al described a pathway that linked nucleolar stress with lipid accumulation in *C.elegans*. It is shown that alteration in ribosome biogenesis (rRNA transcription and processing) leads to increased lipid accumulation and triacylglycerol (TAG) content. They identified the transcription factor PHA-4 that acts as a sensor of nucleolar stress to bind and transactivate the expression of the lipogenic genes *pod-2*, *fasn-1* and *dgat-2*, with consequent increase in lipid biosynthesis and accumulation.

This study initiates with the observation that a mutant for *rrp8* displayed enlarged lipid droplets and increased lipid accumulation and triacylglycerol (TAG) content. In yeast RRP-8 is a nucleolar protein with methyltransferase activity for m1A base modification of 25S rRNA, which was implicated in pre-rRNA cleavage. Its mammalian homolog NML was also reported to modify rRNA. Based on this information the authors described that *rrp8* mutants showed defects in rRNA processing and display a polysome profiles with a decrease in 60S subunits and polysome levels. Interestingly, treatment with Actinomycin D (Act D), an inhibitor of rRNA gene transcription also induced lipid accumulation but did not further enhance lipid accumulation in *rrp-8(kun54)*, suggesting that RRP-8 and Act D act in one pathway, that is ribosome biogenesis. Similarly to *rrp8* mutants, also mutants for *rsks-1*, a homolog of the p70 ribosomal protein S6 kinase and effector of the TOR pathway, showed decreased polysome levels and 60S subunits but also increased lipid accumulation. Because of these results, the authors defined these conditions (*rrp-8* and *rsks-1* mutants and ActD treatment) as nucleolar stress.

The authors then analyzed three factors EGL-9, AAK-2, PHA-4, which were described to mediate lifespan extension of the *rsks-1(ok1255)* mutant. They found that only PHA-4 is required for lipid accumulation observed in *rrp-8* and *rsks-1* mutants and ActD-treated worms. Moreover, PHA-4 expression was increased in *rrp-8* and *rsks-1* mutants and ActD-treated worms. Gene expression analysis revealed that transcription of lipid metabolic genes *fasn-1*, *pod-2* and *dgat-2* was upregulated in *rrp8* and *rsks-1* mutants and ActD-treated worms whereas their expression is suppressed in *pha-4* mutants. The data revealed that PHA-4 associates with the promoter region of *fasn-1*, *pod-2* and *dgat-2* and this association increase in *rrp8* and *rsks-1* mutants. RNAi KD of *fasn-1* and *pod-2* in *rrp-8* and *rsks-1* mutants at L2/L3 developmental stage resulted in low lipid accumulation. Thus, the activities of FASN-1 and POD-2 are required for the increase in lipid droplets in *rrp8* and *rsks-1* mutants. The authors then analyzed DGAT-2, which is described to be required for lipid droplet expansion, and found that it is higher expressed in *rrp8* and *rsks-1* mutants and ActD -treated worms and downregulated upon RNAi knockdown of *pha-4*. Finally, KO of DGAT-2 suppressed lipid accumulation in *rrp8* and *rsks-1* mutants.

The accumulation of liquid droplets upon impairing rRNA transcription by ActD treatment and in *rrp-8* and *rsks-1* mutants is clear, interesting and novel. However, the work as it is remains descriptive and the mechanism by which ribosome biogenesis affects lipogenesis pathway are elusive. The data point to a role of PHA-4 as a nucleolar stress sensor that is required for nucleolar stress-induced lipid accumulation. However, how PHA-4 is regulated by defects of ribosome biogenesis is unclear and not addressed. Moreover, the hypothesis is that under nucleolar stress

excessive energy is stored as fat. Why? Which are the consequences when this storage is inhibited? Finally, the description of how the experiments have been performed is very often lacking important details, making difficult to evaluate the results.

We very much appreciate your comments “*The accumulation of liquid droplets upon impairing rRNA transcription by ActD treatment and in rrp-8 and rsk-1 mutants is clear, interesting and novel*”.

As we know, many intracellular stresses may alter lipid metabolism, such as the well-known ER stress and mitochondria stress. To the best of our knowledge, we may be the first to show that nucleolar stress triggered by systematical disruption of ribosome biogenesis led to lipid accumulation using model organism *C. elegans*. More importantly, we further reveal a new transcription factor PHA-4, not the well-known P53/CEP-1, responses to nucleolar stress to bind to lipogenic genes to promote lipid biosynthesis and accumulation. We think that we have addressed many important issues in this work; but we also agree that many new questions can be raised for future study since the findings in this work are interesting and novel as you mentioned.

Following your nice suggestions, we found that ribosomal protein RPL-11.2 and RPL-5 mediates the activation of PHA-4 under nucleolar stress; nucleolar stress-induced lipid accumulation is required for worm survival under starvation. Below, we will address your suggestions or questions one by one.

Fig. 1

I liked very much how the mutant *kun54* has been analyzed.

Answer:

We added information of analysis of *kun54* mutant as Figure S1 in the revised manuscript. Here, we briefly described the procedure how we did these experiments.

We followed the methodology of SNP (single nucleotide polymorphism) mapping of mutation by *Davis et al.* (Davis et al., 2005), which includes chromosome mapping and interval mapping derived. Forty-eight pairs of designed primers were used to identify the genotype of either Bristol N2 or CB4856 Hawaiian through specific enzyme that digests the corresponding PCR products. The principle of genetic mapping is that SNPs closed to mutation sites would present higher linkage effect.

Firstly, we crossed *kun54* with CB4856 Hawaiian and picked heterozygous F1 whose F2 generation would generate homozygous *kun54* mutant. Secondly, we could separate F2 generation into mutant population (homozygous *kun54*) and non-mutant population (heterozygous *kun54* and wild type). Next, we used these two populations as DNA templates for PCR with 48 pairs of designed primers. PCR products would be digested by specific restriction enzyme to identify the genotype of either Bristol N2 or CB4856 Hawaiian. Since *kun54* mutant was screened from Bristol N2 by EMS mutagenesis, SNPs closed to the mutation site would presents increased tendency of Bristol N2 genotype in the mutant population. Our chromosome mapping showed that the first SNP on chromosome IV was coincident with our hypothesis (Figure S1A).

After we mapped the *kun54* mutation on chromosome IV, we then selected 6 others SNPs near the first SNP (listed in FigureS1B) on this chromosome for interval mapping. Individual *kun54* mutant worms were selected from heterozygous F1 progeny from the cross of *kun54* with CB4856 Hawaiian. Approximate 150 recombinants were selected for interval mapping. As mentioned before, the closer a SNP near to mutation site, the more difficult cross-over occurs. Eventually, we mapped the mutation site of *kun54* in the 0.4 M region on chromosome IV (Figure S1C, D).

At the same time, we also carried out whole genome sequencing of *kun54* mutant. Combined SNP mapping and whole genome sequencing analysis, we were easily able to identify that the mutation of *kun54* is in T07A9.8 (Figure 1F).

In Fig. 1I it is described that RRP8 is ubiquitously expressed in almost all cells throughout all developmental stages. I see more, it appears to me that RRP8 is in the nucleolus. However, this specific localization is lost in Figure S1 due to high exposure. The experiment of Figure S1 is important since it serves to determine that the expression level or nuclear localization of WT and the mutant RRP-8 (G301R)::GFP is the same. A less exposure of this image might have revealed differences in nucleolus/cellular localization between WT and mutant RRP-8 such as a delocalization from the nucleolus.

Answer:

Yes. RRP-8::GFP is indeed localized in nucleolus. Following your excellent suggestion, we confirmed the nucleolus localization of RRP-8::GFP using mCherry::FIB-1, encoding the nucleolar protein fibrillarin. These new data were showed as Figure 1I in the revised manuscript.

In addition, we used confocal microscope to visualize the expression of RRP-8::GFP and RRP-8(G301R)::GFP under same setting but with reduced exposure time. We found that the fluorescence level of RRP-8(G301R)::GFP is obviously lower than that of RRP-8::GFP (Figure S2A in the revised manuscript). Furthermore, in order to examine whether G301R mutation of RRP-8 could alter its localization, we took photos under different sensitivity of detector, but found that the G301R mutation has no effect on the nucleolus localization of RRP-8::GFP (Figure S2B in the revised manuscript).

In Fig. 2 it is described that RRP8 affects rRNA methylation and consequently rRNA processing. The qPCR is a nice method but for analysis of defects in rRNA processing a Northern analysis is an obligatory step. The same is true for the measurements of pre-rRNA. In Fig. 2G the values are detected by qPCR. How have been these data calculated? Short legends such as "Relative rRNA levels detected by qPCR." are not of help.

Answer:

Following your nice suggestion, we used northern blotting to analyze pre-rRNA level and rRNA processing in WT and *rrp-8(kun54)* worms. We designed three probes to detect pre-rRNA level and rRNA intermediates (b, b', c and c') according to previous reports (Figure 2E) (Bousquet-Antonelli et al., 2000; Saijou et al., 2004; Voutev et al., 2006). Digoxin-labeled RNA

probes were generated using DIG northern starter kit (Roche). Similar to our qPCR results, the northern blotting results showed that the pre-rRNA levels were unchanged between wild type and *rrp-8(kun54)* mutant worms. Importantly, we found that the level of band b containing site III, which was specifically recognized by probe 2 (Figure 2E), was also elevated in *rrp-8(kun54)* worms compared with WT worms (Figure 2H), suggesting that the cleavage efficiency of site III corresponding to site A2 in yeast was affected due to dysfunction of RRP-8. These data were consistent with our qPCR results (Figure 2F,G). We added these results to Figure 2E and 2H in the revised manuscript.

Values detected by qPCR are presented as mean $2^{-\text{ddCt}} \pm \text{SD}$ of 4 biologic replicates, using *tbb-2* as an internal control. We added more details to the legends of figure 2 in the revised manuscript.

In Fig. 3 polysome profile data are shown. The decrease in 60S subunit in *rrp-8* and *rsk-1* is for me undetectable. What is relevant here is the decrease in polysomes. Because of defects in rRNA processing described in Fig. 2, one should expect a decrease in 80S levels, which is however not the case. Why? **Moreover, the description of ribosome profile in M&M does not contain the necessary information to evaluate the data.** How have the samples been loaded on the gradient? Same amount of cells, proteins or cytoplasmic RNA? How the peaks have been measured? This is particularly relevant for the very little peak of 60S. I have also some concerns about the treatment with Act D. How long was the treatment? Act D inhibits transcription since is an intercalator with preference for CG-rich sequences. Correct dosage of Act D is key to study rDNA transcription. 50 ng/ml is the concentration of ActD used to analyze rRNA gene transcription since rRNA genes in mammals have a high content of CG. The concentration here used was 15 ng/ μL . Why this concentration? And how long have been treated the worms? This is an important point since elevated ActD concentration inhibits transcription from both Pol II and Pol I genes. If this is the case the ActD treatment is not anymore specific for rRNA genes.

Answer:

We appreciate your excellent and very professional suggestions.

The peak size of 60S subunit in ribosomal profile may be variable in different organisms. According to others reports (Essers et al., 2015; Schosserer et al., 2015), the 60S peak size may be weak generally in *C. elegans*. So it is difficult to detect a significant change of 60S subunit as you pointed out. We removed the description of 60S in the revised manuscript.

From our results, we found that the levels of polysomes were significantly decreased, whereas the levels of 80S were relatively stable in *rrp-8(kun54)*, *rsk-1(ok1255)* and AD treated worms compared with WT worms. These data indicated that the total ribosome levels were significantly decreased and translation was inhibited in the mutants. As we know, polysomes and 80S are different forms of ribosomes. The decreased ribosome biogenesis in *rrp-8(kun54)*, *rsk-1(ok1255)* and AD treated worms was reflected on the descending polysome levels but relatively stable levels of free 80S ribosome. We added the quantification data of 80S to Figure 3C in the revised manuscript.

We added detailed information regarding “Ribosomal profile analysis” to Material and Methods.

In addition, we also added more information of quantification analysis of the polysome and 60S peak size to the legend of Figure 3C in the revised manuscript. Please also see below text.

600 μ l homogenated sample (normalized to 260nm absorbance, equal amount of cytosolic RNA) was applied to the top of sucrose gradients for ultracentrifugation at 36,000 rpm, 3 hours, 4°C. Gradients were fractionated with continuous monitoring from bottom to top by 254nm absorbance. The areas under each peak were calculated using ImageJ software. The fold-change of each peak size of a specific worm strain is presented compared with the 60S peak size of wild type (WT). Data are presented as the means \pm SD of 3 biological repeats. This part complementary information was included in the figure legend of the revised manuscript.

We modified “**Actinomycin D (AD) treatment**” in **Extended experimental procedures of Supplementary information**.

Please see below information regarding that we used 15 ng/ μ L AD in this study.

C. elegans are raised on the surface of NGM (nematode growth medium) with *E.coli* OP50 as food. We added actinomycin D (AD) to NGM plates at the final concentration 15 ng/ μ L. L1 stage worms were seeded on the plate supplemented with or without AD, and developed to one-day of adulthood for different experiment analysis. As you mentioned, 50 ng/ml is the concentration of ActD often used to analyze rRNA gene transcription in the cell systems. We know, cells are cultured in the liquid medium and are sensitive to treatment of compounds. It may be understandable that the concentration of AD for cells treatment may be different in worms. In order to explore what concentration of AD could induce lipid accumulation in the worms when we started this work, we designed several gradient concentrations of AD to treat wild type worms. As showed in Response Figure 9, we found that, compared with the higher concentrations of AD, 15 ng/ μ L AD treatment showed significantly increased lipid accumulation but nearly no obvious effect on development in worms. These preliminary experiments made us decide to use 15 ng/ μ L as the concentration of AD for this study.

Response Figure 9. The effects of AD concentrations on development/growth and lipid accumulation in wild type (WT) worm.

The statement "suggesting that Actinomycin D plays an evolutionarily conserved function to inhibit rDNA transcription from worms to mammals" should be toned down. First, no data on rRNA gene transcription are shown. Second, a chemical compound cannot play an evolutionary conserved function.

Answer:

We thank and agree to your suggestion. We removed this sentence in the revised manuscript.

In Fig. 4 it is described the identification of PHA-4. The data clearly showed that the formation of lipid droplets in *rrp-8* and *rsks-1* mutants and ActD treated worms require the expression of PHA-4. However, there is a lack of information how the experiments have been performed. It is described that "The temperature-sensitive strain *pha-4(zu225);smg-1(cc546ts)* showed inactivation of PHA-4 under a restrictive temperature of 20°C". No data have been shown on this or reported through reference.

Answer:

According to previous report by Dr. Susan Mango's laboratory (Updike and Mango, 2007), *SM190[pha-4(zu225);smg-1(cc546ts)]* is a worm strain containing two genes mutation. Temperature sensitive mutant *smg-1(cc546ts)*, which is the component of nonsense-mediated decay (NMD) pathway, exhibits robust NMD activity at 15 °C, but compromises activity at higher

temperature. At restrictive temperature 15 °C, high NMD activity of SMG-1(cc546) degrades *pha-4* mRNA from *pha-4(zu225)*, leading to lethality of *SM190* worms at L1 stage; while at 24 °C, SMG-1(cc546ts) was compromised and *pha-4* mRNA is stabilized and accumulated, leading to survival of *SM190* worms.

We cited the reference by Updike and Mango in the revised manuscript. All double and triple mutants that contain *SM190[pha-4(zu225);smg-1(cc546ts)]* background were raised at 24 °C, and their synchronized eggs were hatched to L1 at 24 °C. And then these synchronized L1 worms were placed onto NGM plates and cultured at 20 °C for experiments.

In order to explore whether *pha-4(zu225)* mutant could suppress the increased lipid accumulation in *rrp-8(kun54)*, *rsks-1(ok1255)* and AD treated worms, we crossed the *rrp-8(kun54)* and *rsks-1(ok1255)* into *SM190 [pha-4(zu225);smg-1(cc546ts)]*, respectively, at 24 °C. We can easily identify the mutation background of *rrp-8(kun54)*, *rsks-1(ok1255)*, or *pha-4(zu225)* through PCR and sequencing. The exact mutation site of *smg-1(cc546ts)* has not been annotated by CGC and Wormbase. In order to identify the homozygous *smg-1(cc546ts)*, we examined about 10 lines of each double mutants *rrp-8(kun54);pha-4(zu225)* and *rsks-1(ok1255);pha-4(zu225)* that might contain *smg-1(cc546ts)* background. We raised these lines of mutant worms at 24 °C for normal reproduction. Their progenies were raised at 15 °C to identify the activity of SMG-1(cc546ts). A line of worms, which were lethal at 15 °C, indicated that they contained robust nonsense-mediated decay (NMD) activity, and identified as *rrp-8(kun54);SM190[pha-4(zu225);smg-1(cc546ts)]* and *rsks-1(ok1255);SM190[pha-4(zu225);smg-1(cc546ts)]*.

We added these detailed information to Extended experimental procedures in Supplemental information in the revised manuscript.

Data of Fig. 4C showing % TAG in lipids in the *smg-1(cc546ts)* mutation alone should be integrated in Fig. 4D or at least use the same Y axis scale. At which temperature the analysis of the *smg-1(cc546ts)* control has been performed?

Answer:

We adjusted the Y axis scale in Figure 4C. Like all of other mutants, the *smg-1(cc546ts)* control was also raised at 20 °C in our experiments.

The increase expression of PHA-4 mRNA in *rrp-8* and *rsks-1* mutants and ActD treated worms is very clear. However, I do not see any dramatic increase in "fluorescence of PHA-4::GFP (*unc-119(ed3);wgIs37*) (Fig.4E)". In contrast, it appears that cells are enlarged. The western in Fig. 4F lacks clarity and the short legend "Immunoblotting of PHA-4::GFP with anti-GFP antibody." does not help. What are a, b and c? How the valued of fold expression have been calculated? Why are there so many bands?

Answer:

We replaced "dramatic" with "obviously" in the revised manuscript.

Please also see our response to similar comments by Reviewer #2.

Based on information provided by Wormbase (Response Figure 6), *pha-4* can be transcribed into three isoforms a (the longest one), b (the middle one), c (the shortest one). To clarify which isoform could respond to nucleolar stress, we generated three constructs *pha-4a::GFP*, *pha-4b::GFP* and *pha-4c::GFP* using individual transcript of *pha-4* to fuse to GFP. We found that, similar to PHA-4::GFP (*wgIs37[pha-4::GFP]*) (Figure 4F), PHA-4a::GFP, PHA-4b::GFP and PHA-4c::GFP all express in nuclei (Response Figure 7). Based on western blot analysis, we could judge that PHA-4a::GFP is the primary band upregulated in *rrp-8(kun54)*, *rsks-1(ok1255)* and AD treated animals compared with wild type (WT) (Replicates 1, 2, 3, and 4 in Response Figure 8. Replicate 2 was shown in the initial submitted manuscript). The band sizes of PHA-4b::GFP and PHA-4c::GFP are very close, and also shorter than that of PHA-4a::GFP. It is pretty hard to detect PHA-4b::GFP and PHA-4c::GFP levels through several times of western blotting (Replicates 1 and 2 in Response Figure 8). Consistently, fluorescence microscopy also showed that PHA-4b::GFP and PHA-4c::GFP expression levels were obviously lower than PHA-4a::GFP (Response Figure 7). Therefore, we did not add PHA-4b::GFP and PHA-4c::GFP to Figure 4G in the revised manuscript.

In order to minimize the interference of unspecific bands, we optimized western blotting by prolonging blocking time (overnight) and also used new diluted anti-GFP antibody for immunoblotting (Replicate 3 and 4). In addition, we used WT worms without GFP as a negative control, and WT worms with PHA-4a::GFP *{kunEx136[Pvha-6::pha-4a::GFP]}* as a positive control (Replicate 3 and 4, Response Figure 8). Consistently, the expression levels of PHA-4::GFP is obviously increased in *rrp-8(kun54)*, *rsks-1(ok1255)* and AD treated animals compared with WT animals. We replaced Figure 4G with the result of Replicate 4 in the revised manuscript.

I will also be careful in the normalization considering that the defects observed in polysome profiles might affect the total amounts of proteins (see the 4.4 fold change in the ActD sample). Finally, it is indicated that "PHA-4::GFP is expressed in the nucleus of intestine cells, similar to nuclear expression of RRP-8::GFP. What is relevant here? The nuclear expression or the expression in intestine cells? Please clarify.

Answer:

We agree with your cautious consideration on these data. We used the level of actin protein and the amount of total proteins as internal control in western blot. Generally, we loaded equal amount proteins (30 µg) of each sample for gel running.

PHA-4::GFP expresses in many types of cells, such as pharyngeal and intestine cells, in *C. elegans*. RRP-8::GFP ubiquitously expressed in almost all cells throughout all developmental stages from embryo to adulthood (Figure 1Ja-f).

We are sorry for making confusion of PHA-4::GFP and RRP-8::GFP expression pattern. That should be the expression in intestine cells. We removed these sentences in the revised manuscript.

What is unclear here is how PHA-4 get activated, an issue that is ignored in the whole manuscript.

Answer:

As reported previously, p53-dependent or independent nucleolar stress pathways are almost all mediated by ribosomal proteins RPL11 and RPL5, which may have the function in gene regulation (Donati et al., 2011; Kuroda et al., 2011). RPL11/RPL5 are necessary for nucleolar stress pathways, since depletion of RPL11/RPL5 impairs p53-dependent or independent nucleolar stress response (Fumagalli et al., 2009; Sundqvist et al., 2009). We wondered whether *pha-4* upregulation under nucleolar stress was also dependent on RPL11 or RPL5 in *C. elegans*. *rpl-11.1* and *rpl-11.2* are two orthologs of mammalian *rpl11*, and *rpl-5* is an ortholog of mammalian *rpl5* in *C. elegans*. We found that knockdown of *rpl-11.1*, *rpl-11.2* and *rpl-5* in WT and *rrp-8(kun54)* expressing PHA-4::GFP shown that *rpl-11.2 RNAi* and *rpl-5 RNAi* would significantly suppressed upregulation of PHA-4::GFP in *rrp-8(kun54)* (Figure 5A and 5B in the revised manuscript). Furthermore knockdown of *rpl-11.2* and *rpl-5* also decreased the fat contents in *rrp-8(kun54)* and *rsks-1(ok1255)* similar to *pha-4* mutation (Figure 5D in the revised manuscript). These results indicate that the function of RPL11/RPL5 in response to nucleolar stress is evolutionarily conserved in *C. elegans* and mammals

In Fig. 5, the analysis of RNAseq in *rrp8* mutants revealed the activation of *dgat-2*, *fasn-1* and *pod-2* genes, which are implicated in lipogenesis. Expression of these genes depends on PHA-4 levels. The PHA-4 ChIP analysis indicated PHA4 bind the promoter of these genes. I would like to see here further controls to substantiate the specificity of this assay such as amplifications of genes with transcription not affected by *rrp8*, *rsks-1* and AD treatment (qRT-PCR, Fig. 5B) and sequences not bound by PHA-4 (ChIP, Fig. 5E-F).

Answer:

Following your nice suggestion, we tested the promoters of two genes *taf-1* and *myo-2* as the negative and positive control of ChIP-QPCR, respectively (Figure S7A). *taf-1* was previously reported as a negative control of PHA-4::GFP (Hsu et al., 2015), and its expression was unchanged in our transcriptome of wild type and *rrp-8(kun54)*.

In addition, we mutated the predicted binding sites (*m1*, *m2*) of *dgat-2* promoter to confirm the specificity binding of PHA-4 in the revised manuscript (Figure S7B-D).

PHA-4 binding sites.

Our ChIP-qPCR analyses showed PHA-4 has ability to bind to the promoter regions of *fasn-1*, *pod-2* and *dgat-2* (Figure S7A), and the binding of PHA-4 to these genes showed an increased tendency in both the *rrp-8(kun54)* and *rsks-1(ok1255)* mutants compared with that in WT worms (Figure 6D-F). To test the binding matters of for induction of these target genes, we opted to investigate the expression of *dgat-2*, since it involves in the last step of triacylglycerol (TAG) biosynthesis and its promoter has the minimum number of putative binding sites for PHA-4 compared with *fasn-1* and *pod-2* (Figure 6C). We generated four constructs with the full-length promoter (*full*) or truncated promoters (*m1*, *m2*, *m1+m2*) of *dgat-2* fused to green fluorescent protein (GFP) as a reporter (Figure S7B). We found that the *m2*, but not the *m1* binding site of *dgat-2* promoter is crucial for its induction. These experiments including ChIP-qPCR analysis consistently show that *dgat-2* is a direct target of PHA-4, and PHA-4 can directly binds to the *m2*

element to transactivate its expression under nucleolar stress. These data were added to revised manuscript as Figure S7B-D.

In Fig. 6 it is described DGAT-2. The data in Fig. 6A are very clear and support the data of Fig. 5 showing that DGA2 expression depends of PHA-4 expression. It is however not a surprise that deletion of DGAT-2 impairs accumulation of lipid droplets in in *rrp-8* and *rsks-1* mutants and ActD treated worms. Indeed, as indicated by the authors DGAT-2 catalyzes the conjugation of a fatty acyl-CoA to diacylglycerol (DAG) to form TAG and is a lipid droplet (LD) protein required for LD expansion. It might have been interesting here to determine what happens to the lipid droplets through KD of another gene implicated in lipogenesis but with expression not altered upon ribosome biogenesis (i.e. *rrp8* mutants).

Answer:

Yes. We actually used such a gene, *sbp-1* in our work. The transcription factors sterol regulatory element binding proteins (SREBPs) is a well-known master regulator of lipogenesis from *C. elegans* to mammals. SREBPs play essential roles to regulate the biosynthesis of fatty acids, triglycerides, and cholesterol to meet the needs of the cell (Goldstein et al., 2006; Horton et al., 2002; Jeon and Osborne, 2012; Raghow et al., 2008; Shao and Espenshade, 2012). The activities and functions of SREBPs in lipid metabolism are highly evolutionarily conserved across metazoans. The model organism *C. elegans* contains only one SREBP family member encoded by *sbp-1*. We and other *C. elegans* laboratories have shown that, similar to its mammalian homologues, SBP-1 also regulates fatty acid (Kniazeva et al., 2008; Liang et al., 2010; McKay et al., 2003; Nomura et al., 2010), phosphatidylcholine (PC) (Walker et al., 2011), and cholesterol metabolism (Walker et al., 2010; Yang et al., 2006).

In this work, we showed that The fluorescence intensity of GFP::SBP-1 (*ftIs7[Psbp-1::gfp::sbp-1]*) was indistinguishable between the *rrp-8(kun54)* mutant and WT worms (Figure S4A). Meanwhile, the TAG content is still increased in *sbp-1(ep79)* mutant background (Figure S4C, D). In addition, following your nice suggestion, we further tested the expression of *pod-2* encoding acetyl-CoA carboxylase, *fasn-1* encoding fatty acid synthase and *dgat-2* encoding diacylglycerol O-acyltransferase 2. We found that the mRNA expression of these genes were still upregulated in *rrp-8(kun54);sbp-1(ep79)* mutants background compared with *sbp-1(ep79)* mutant background. Altogether, these data consistently demonstrate that nucleolar stress-induced lipid accumulation is independent on transcription factor SBP-1/SREBP.

Response Figure 10. Relative mRNA expression of *dgat-2*, *fasn-1* and *pod-2* in wild type (Con) and *sbp-1(ep79)* backgrounds.

Finally, which are the consequences to abolish lipid droplets upon "nucleolar stress"?

Answer:

Please also see our above response to Reviewer #1.

Many studies have shown that animals/cells often shift their metabolism to increased nutrient storage and efficiency of energy utilization under harsh conditions like prolonged starvation. The nucleolus is a central hub for coordinating the stress response, since nearly all metabolic and signaling pathways ultimately lead to or from it.

Our present study found that nucleolar stress promotes lipid accumulation; we consequently thought whether nucleolar stress can help worm to live longer under starvation, in order to address this question. We then performed starvation experiments by leaving synchronized L4 worms in M9 buffer without food, and counting living animals every day. Indeed, we found that *rrp-8(kun54)* and *rsk-1(ok1255)* mutant worms, as well as AD treated worms, displayed extended survival compared with wild type (WT) worms under starvation condition, suggesting that nucleolar stress is beneficial for worms survival. Meanwhile, we found that the increased survival depends on excessive lipid accumulation because mutation of the *dgat-2* abolished this benefit. We added these results as Figure S10 in the revised manuscript.

During the revision, we found a report by Dr. Alexander Soukas's group. They also showed that reduced function of cytoplasmic aminoacyl tRNA synthetases (ARS genes) leads to increased fat storage and extended starvation survival in *C. elegans*. Consistently, a very early report also showed that genetic activation of protein synthesis results in opposite effects in *Drosophila*. Therefore, metabolic alteration via reduced ribosome biogenesis or protein production may indeed help animals to survive under starvation condition.

Minor points

The analyses of *cep1* mutants in lipid formation are in Fig. S2C and S2D (and not S3C and S4D).

Answer:

Sorry for this mistake. We corrected it in the revised manuscript.

Fig 2C represents 6n values which are not shown in Fig. 2B. 80S values should be shown.

Answer:

We added 80S value to Figure 3C in the revised manuscript.

References for Responses.

- Avery, L. (1993). The genetics of feeding in *Caenorhabditis elegans*. *Genetics* *133*, 897-917.
- Bousquet-Antonelli, C., Vanrobays, E., Gelugne, J.P., Caizergues-Ferrer, M., and Henry, Y. (2000). Rrp8p is a yeast nucleolar protein functionally linked to Gar1p and involved in pre-rRNA cleavage at site A2. *RNA (New York, NY)* *6*, 826-843.
- Brooks, K.K., Liang, B., and Watts, J.L. (2009). The influence of bacterial diet on fat storage in *C. elegans*. *PLoS One* *4*, e7545.
- Davis, M.W., Hammarlund, M., Harrach, T., Hullett, P., Olsen, S., and Jorgensen, E.M. (2005). Rapid single nucleotide polymorphism mapping in *C. elegans*. *BMC genomics* *6*, 118.
- Donati, G., Brighenti, E., Vici, M., Mazzini, G., Trere, D., Montanaro, L., and Derenzini, M. (2011). Selective inhibition of rRNA transcription downregulates E2F-1: a new p53-independent mechanism linking cell growth to cell proliferation. *Journal of cell science* *124*, 3017-3028.
- Essers, P.B., Nonnekens, J., Goos, Y.J., Betist, M.C., Viester, M.D., Mossink, B., Lansu, N., Korswagen, H.C., Jelier, R., Brenkman, A.B., *et al.* (2015). A Long Noncoding RNA on the Ribosome Is Required for Lifespan Extension. *Cell reports*.
- Friedman, J.R., and Kaestner, K.H. (2006). The Foxa family of transcription factors in development and metabolism. *Cellular and molecular life sciences : CMLS* *63*, 2317-2328.
- Fumagalli, S., Di Cara, A., Neb-Gulati, A., Natt, F., Schwemberger, S., Hall, J., Babcock, G.F., Bernardi, R., Pandolfi, P.P., and Thomas, G. (2009). Absence of nucleolar disruption after impairment of 40S ribosome biogenesis reveals an rpL11-translation-dependent mechanism of p53 induction. *Nat Cell Biol* *11*, 501-508.
- Goldstein, J.L., DeBose-Boyd, R.A., and Brown, M.S. (2006). Protein sensors for membrane sterols. *Cell* *124*, 35-46.
- Horton, J.D., Goldstein, J.L., and Brown, M.S. (2002). SREBPs: activators of the complete program of cholesterol and fatty acid synthesis in the liver. *J Clin Invest* *109*, 1125-1131.
- Hsu, H.T., Chen, H.M., Yang, Z., Wang, J., Lee, N.K., Burger, A., Zaret, K., Liu, T., Levine, E., and Mango, S.E. (2015). TRANSCRIPTION. Recruitment of RNA polymerase II by the pioneer transcription factor PHA-4. *Science* *348*, 1372-1376.
- Jeon, T.I., and Osborne, T.F. (2012). SREBPs: metabolic integrators in physiology and metabolism. *Trends Endocrinol Metab* *23*, 65-72.
- Kniazeva, M., Euler, T., and Han, M. (2008). A branched-chain fatty acid is involved in post-embryonic growth control in parallel to the insulin receptor pathway and its biosynthesis is feedback-regulated in *C. elegans*. *Genes Dev* *22*, 2102-2110.
- Kuroda, T., Murayama, A., Katagiri, N., Ohta, Y.M., Fujita, E., Masumoto, H., Ema, M., Takahashi, S., Kimura, K., and Yanagisawa, J. (2011). RNA content in the nucleolus alters p53 acetylation via MYBBP1A. *EMBO J* *30*, 1054-1066.
- Lakowski, B., and Hekimi, S. (1998). The genetics of caloric restriction in *Caenorhabditis elegans*. *Proc Natl Acad Sci U S A* *95*, 13091-13096.
- Liang, B., Ferguson, K., Kadyk, L., and Watts, J.L. (2010). The role of nuclear receptor NHR-64 in fat storage regulation in *Caenorhabditis elegans*. *PLoS One* *5*, e9869.
- Mango, S.E., Lambie, E.J., and Kimble, J. (1994). The Pha-4 Gene Is Required to Generate the Pharyngeal Primordium of *Caenorhabditis-Elegans*. *Development* *120*, 3019-3031.
- McKay, R.M., McKay, J.P., Avery, L., and Graff, J.M. (2003). *C. elegans*: a model for exploring the genetics

of fat storage. *Dev Cell* 4, 131-142.

Na, H., Zhang, P., Chen, Y., Zhu, X., Liu, Y., Xie, K., Xu, N., Yang, F., Yu, Y., Cichello, S., *et al.* (2015). Identification of lipid droplet structure-like/resident proteins in *Caenorhabditis elegans*. *Biochim Biophys Acta*.

Nomura, T., Horikawa, M., Shimamura, S., Hashimoto, T., and Sakamoto, K. (2010). Fat accumulation in *Caenorhabditis elegans* is mediated by SREBP homolog SBP-1. *Genes Nutr* 5, 17-27.

Panowski, S.H., Wolff, S., Aguilaniu, H., Durieux, J., and Dillin, A. (2007). PHA-4/Foxa mediates diet-restriction-induced longevity of *C. elegans*. *Nature* 447, 550-555.

Raghow, R., Yellaturu, C., Deng, X., Park, E.A., and Elam, M.B. (2008). SREBPs: the crossroads of physiological and pathological lipid homeostasis. *Trends Endocrinol Metab* 19, 65-73.

Saijou, E., Fujiwara, T., Suzaki, T., Inoue, K., and Sakamoto, H. (2004). RBD-1, a nucleolar RNA-binding protein, is essential for *Caenorhabditis elegans* early development through 18S ribosomal RNA processing. *Nucleic acids research* 32, 1028-1036.

Schossere, M., Minois, N., Angerer, T.B., Amring, M., Dellago, H., Harreither, E., Calle-Perez, A., Pircher, A., Gerstl, M.P., Pfeifenberger, S., *et al.* (2015). Methylation of ribosomal RNA by NSUN5 is a conserved mechanism modulating organismal lifespan. *Nature communications* 6, 6158.

Shao, W., and Espenshade, P.J. (2012). Expanding roles for SREBP in metabolism. *Cell Metab* 16, 414-419.

Sundqvist, A., Liu, G., Mirsaliotis, A., and Xirodimas, D.P. (2009). Regulation of nucleolar signalling to p53 through NEDDylation of L11. *EMBO reports* 10, 1132-1139.

Urdike, D.L., and Mango, S.E. (2007). Genetic suppressors of *Caenorhabditis elegans* pha-4/FoxA identify the predicted AAA helicase *ruvb-1/RuvB*. *Genetics* 177, 819-833.

Voutev, R., Killian, D.J., Ahn, J.H., and Hubbard, E.J. (2006). Alterations in ribosome biogenesis cause specific defects in *C. elegans* hermaphrodite gonadogenesis. *Developmental biology* 298, 45-58.

Walker, A.K., Jacobs, R.L., Watts, J.L., Rottiers, V., Jiang, K., Finnegan, D.M., Shioda, T., Hansen, M., Yang, F., Niebergall, L.J., *et al.* (2011). A conserved SREBP-1/phosphatidylcholine feedback circuit regulates lipogenesis in metazoans. *Cell* 147, 840-852.

Walker, A.K., Yang, F., Jiang, K., Ji, J.Y., Watts, J.L., Purushotham, A., Boss, O., Hirsch, M.L., Ribich, S., Smith, J.J., *et al.* (2010). Conserved role of SIRT1 orthologs in fasting-dependent inhibition of the lipid/cholesterol regulator SREBP. *Genes Dev* 24, 1403-1417.

Yang, F., Vought, B.W., Satterlee, J.S., Walker, A.K., Jim Sun, Z.Y., Watts, J.L., DeBeaumont, R., Saito, R.M., Hyberts, S.G., Yang, S., *et al.* (2006). An ARC/Mediator subunit required for SREBP control of cholesterol and lipid homeostasis. *Nature* 442, 700-704.

Yen, K., Le, T.T., Bansal, A., Narasimhan, S.D., Cheng, J.X., and Tissenbaum, H.A. (2010). A comparative study of fat storage quantitation in nematode *Caenorhabditis elegans* using label and label-free methods. *PLoS One* 5.

Zhang, P., Na, H., Liu, Z., Zhang, S., Xue, P., Chen, Y., Pu, J., Peng, G., Huang, X., Yang, F., *et al.* (2012). Proteomic study and marker protein identification of *Caenorhabditis elegans* lipid droplets. *Mol Cell Proteomics* 11, 317-328.

Zhang, S.O., Trimble, R., Guo, F., and Mak, H.Y. (2010). Lipid droplets as ubiquitous fat storage organelles in *C. elegans*. *BMC Cell Biol* 11, 96.

Reviewer #1 (Remarks to the Author):

This is a resubmission from Wu et al examining the response to low ribosomes/nucleolar stress in worms and the idea that this promotes survival in times of starvation. The data are intriguing but the authors need really good controls for their experiments – in part because some of the effects are subtle (droplet size, TAG) and in part because some experiments have no controls at all. I ask the authors to do this properly, for example (enclosed in asterisks are particularly critical):

- Figure 2: quantify. And show positive controls for quantitation e.g. that RNA levels loaded were equivalent, for example, and not degraded – with a second probe that is not rRNA.
- Nucleolar stress triggers lipid accumulation: actinomycin D is not very specific, as an intercalator in DNA and RNA, but at low concentrations should mostly target RNAP I. See Bensaude 2011. What controls show that Act D is specific and not disrupting a lot of processes.
- Figure 2: ribosomal profiles. What are the blue and green lines, how were they set? Is it fair to set mutants to wildtype levels – that is, how variable is the wildtype, and how can the authors normalize the mutants to wildtype in a convincing way. For example, *what is the ratio of polysomes vs 80S ribosome in the different samples*. This might be more compelling.
- Figure 4D: the comparison for TAG should leave out the ### comparison. It should focus on the induction by rrp or rsk in a wild-type background vs the induction in the SM190 background. The effects are rather small, however.
- *Figure 4: induction of PHA-4 is not convincing as shown.* For example, in the western blots in part G, actin increases also so the quantitation should be calculated. In the section F, it is hard to see an increase in GFP, things look about the same and are not quantified. Section E has no normalization for sample consistency. Figure 5 has a similar issue. The worms are hard to visualize in some panels.
- Figures 6 and 7 were more convincing e.g the GFP assays A-E.
- Survival: The data in S10 are intriguing. And suggest an exciting result, but the result is preliminary.* It appears to be an n=1? There should be a table and better statistics.*

In addition some caveats should be mentioned in the paper, for example:

- Eat-2 data should be put in Results, it is a critical control. However, the pha-4 mutant at 20° is stronger than the eat-2 mutant because pha-4 mutants will arrest. This caveat should be spelled out in the paper.

Stylistic:

- allele numbers are not a good way to label figures. It is confusing to readers.

- If space is an issue, some of Figure 2 could be moved to supplemental.

Reviewer #2 (Remarks to the Author):

The authors have addressed most of my concerns. The following points should be further clarified.

1. Abstract: "inactivation of PHA-4 and DGAT-2 is sufficient to abolish nucleolar stress-induced lipid accumulation". Shouldn't it be "inactivation of PHA-4 or DGAT-2"?

2. The following statement is not entirely supported by experimental data. "Interestingly, the G301R mutation of RRP-8 showed an obvious decrease in expression (Figure S2A)". It is unclear if multiple independent transgenic lines were examined. Changes in expression levels may also be due to different number of copies of DNA constructs that were incorporated in the extra-chromosomal arrays.

3. It remains confusing if the 200x or 400x images in Figure 7A were used to yield the quantitative data in Figure 7B. Since the authors conceded that the 200x images were complicated by autofluorescence, they should be removed.

Reviewer #3 (Remarks to the Author):

The authors have made a significant effort to address reviewer's comments, clarify data presentation and add new data that clearly supports their main findings. I believe that the manuscript is now fit for publication.

Response to reviewers:

Reviewer #1 (Remarks to the Author):

This is a resubmission from Wu et al examining the response to low ribosomes/nucleolar stress in worms and the idea that this promotes survival in times of starvation. The data are intriguing but the authors need really good controls for their experiments – in part because some of the effects are subtle (droplet size, TAG) and in part because some experiments have no controls at all. I ask the authors to do this properly, for example (enclosed in asterisks are particularly critical):

We very much appreciate your nice suggestions, and address these issues one by one as below Responses.

- Figure 2: quantify. And show positive controls for quantitation e.g. that RNA levels loaded were equivalent, for example, and not degraded – with a second probe that is not rRNA.

Response:

Following your nice suggestion, we used both *act-1* and *tbb-2* as controls for northern blotting. We loaded equal amounts of total RNA (5 µg), which were from the remaining mRNA samples of our previous experiments in Figure 2H, on a 1.2% denaturing formaldehyde/agarose gel, and then analyzed the mRNA levels of *act-1* and *tbb-2* by northern blotting. The experiment information was added to figure legend of the revised Figure 2H. In addition, we performed northern blotting to analyze the mRNA level of *act-1* and *tbb-2* as internal controls. Like the previous results of Figure 2H and also below Response Figure 11, the quality of total RNA samples was pretty good and did not show degradation. Meanwhile, the northern blotting results showed a clear dark band of either *act-1* or *tbb-2*. We added the result of *tbb-2* northern blotting in the Figure 2H in the revised manuscript.

Response Figure 11. Northern blotting of *act-1* and *tbb-2* with the digoxin-labeled antisense RNA probes, respectively.

- Nucleolar stress triggers lipid accumulation: actinomycin D is not very specific, as an intercalator in DNA and RNA, but at low concentrations should mostly target RNAP I. See Bensaude 2011. What controls show that Act D is specific and not disrupting a lot of processes.

Response:

Yes, actinomycin D (AD) preferentially inhibits the rRNA production by intercalating into GC-rich region of rDNA to inhibit the RNA polymerase I (RNAP I) mediated transcription. AD is generally accepted as specifically inhibiting the RNAP I transcription that can induce nucleolar stress in cells¹⁻⁴. As Bensaude reported, class I gene transcription is the most sensitive under AD treatment although all three polymerases mediated transcription is also affected⁵.

In our work, we showed that AD treated worms displayed decreased ribosomal profile (Figure 3B), increased level of *CEP-1/P53* protein (Supplementary Fig. 3), which has been widely reported as a sensor of nucleolar stress, nearly no obvious effect on development (Response Figure 9). Importantly, we showed for the first time that, similar to the results of genes mutation involved in ribosome biogenesis, AD treatment did lead to lipid accumulation in worms.

Agreed with the reviewer's concern, we also wondered whether AD may affect a lot of other processes. Therefore, we tested whether AD treatment could induce other intracellular stresses, in addition to nucleolar stress. However, we did not find that AD treatment could cause ER stress, mitochondria stress, and also cytosolic stress. As showed in Response Figure 12, AD had no effect on ER unfolded protein response indicated by expression of *Phsp-4::GFP*, a reporter for ER UPR; mitochondria

unfolded protein response indicated by expression of *Phsp-6::GFP*, a marker for mitochondria UPR; and also cytosolic heat shock response (HSR) indicated by expression of *Phsp-16.2::GFP*, a marker for cytosolic HSR. Reversely, these results may be a good control for AD which is specific to nucleolar stress.

Response Figure 12. The effects of actinomycin D (AD) on ER UPR, mitochondria

UPR and cytosolic HSR. (A) *Phsp-4::GFP* reporter worms treated with AD or tunicamycin as a positive control of ER UPR (25 ng/μl, 4 hrs⁶). (B) *Phsp-6::GFP* reporter worms treated with AD or RNAi of *cco-1* as a positive control of mitochondria UPR⁷. (C) *Phsp-16.2::GFP* reporter worms treated with AD or heat shock for 6 hr at 31 °C as a positive control of cytosolic HSR⁸.

- Figure 2: ribosomal profiles. What are the blue and green lines, how were they set? Is it fair to set mutants to wildtype levels – that is, how variable is the wildtype, and how can the authors normalize the mutants to wildtype in a convincing way. For example, *what is the ratio of polysomes vs 80S ribosome in the different samples*. This might be more compelling.

Response:

The blue and green lines in the Figure 3B were two auxiliary lines we added in order to better visualize and distinguish the difference between groups of polysomes. We set the blue line according to the 60S peak point of wild type ribosomal profile, and the green line according to the 2n polysome peak point of wild type ribosomal profile.

Agreed with the reviewer's concern, we also considered this question when we performed the ribosomal profile analysis previously. In order to resolve this problem, homogenated samples were adjusted to equal A260 absorbance for ultracentrifugation. In other words, we had normalized these ribosomal profiles to cytosolic RNA to control the samples consistency. Certainly, as you suggested, the ratio of polysomes vs 80S ribosome in different samples may be more compelling and convincing for the quantification. Therefore, we made the change of Figure 3C in the revised manuscript following your suggestions.

- Figure 4D: the comparison for TAG should leave out the ### comparison. It should focus on the induction by *rrp* or *rsks* in a wild-type background vs the induction in the SM190 background. The effects are rather small, however.

Response:

We thank the reviewer's comment. We removed the ### comparison in the revised manuscript.

- *Figure 4: induction of PHA-4 is not convincing as shown.* For example, in the western blots in part G, actin increases also so the quantitation should be calculated. In the section F, it is hard to see an increase in GFP, things look about the same and are not quantified. Section E has no normalization for sample consistency. Figure 5 has a similar issue. The worms are hard to visualize in some panels.

Response:

We have performed some replicates for western blot of these groups, and all consistently showed an elevated protein level of PHA-4 under nucleolar stress

(Response Figure 8). We quantified the GFP band by normalization to Actin and added the relative protein levels in Figure 4H in the revised manuscript.

As you suggested, we added the quantification of PHA-4::GFP fluorescence intensity in Figure 4G in the revised manuscript. Indeed, the relative levels of GFP intensity in *rrp-8(kun54)*, *rsk-1(ok1255)* and AD treated worms were significantly increased compared with wild type animals (Figure 4G). Consistently, the relative mRNA levels of *pha-4*, which was normalized to *tbb-2* as an internal control, in these worm groups were also elevated compared with wild type animals (Figure 4E).

In addition, we also submitted much higher resolution Figures to avoid hard visualization in some panels in the revised manuscript.

- Figures 6 and 7 were more convincing e.g the GFP assays A-E.

Response: We thank the reviewer's comment.

- Survival: The data in S10 are intriguing. And suggest an exciting result, but the result is preliminary.* It appears to be an n=1? There should be a table and better statistics.*

Response:

We performed 4 biological replicates to count the number of live worms under starvation. In each replicate, 30 L4 stage worms unless specifically indicated, were transferred to M9 buffer without food to assay starvation survival. So, a total of 120 animals of each worm strain were used for starvation survival assay and summarized in Supplementary Fig. 10. We also added the information in the legend of Supplementary Fig. 10. Following the reviewer's useful suggestion, we further carried out statistical analysis of starvation survival data. We calculated mean survival time, median survival time and 75% th survival time to present the difference in each worm strains. The statistical analysis of starvation survival was listed in the Supplementary Table 1 in the revised manuscript.

In addition some caveats should be mentioned in the paper, for example:

- Eat-2 data should be put in Results, it is a critical control. However, the *pha-4* mutant at 20° is stronger than the *eat-2* mutant because *pha-4* mutants will arrest. This caveat should be spelled out in the paper.

Response:

We had already added the *eat-2* data (please see the text in Line 269-277, and Supplementary Fig. 6) in Results. We also mentioned the caveat about *eat-2* mutant and *pha-4* mutant in the revised manuscript.

Stylistic:

- allele numbers are not a good way to label figures. It is confusing to readers.
- If space is an issue, some of Figure 2 could be moved to supplemental.

Response:

We are sorry for the confusing to readers. We modified and used the gene name to label all figures in the revised manuscript. We thank you for your suggestion on Figure 2. There is enough space for all subfigures.

Reviewer #2 (Remarks to the Author):

The authors have addressed most of my concerns. The following points should be further clarified.

1. Abstract: “inactivation of PHA-4 and DGAT-2 is sufficient to abolish nucleolar stress-induced lipid accumulation”. Shouldn't it be “inactivation of PHA-4 or DGAT-2”?

Response: We corrected it.

2. The following statement is not entirely supported by experimental data. “Interestingly, the G301R mutation of RRP-8 showed an obvious decrease in expression (Figure S2A)”. It is unclear if multiple independent transgenic lines were examined. Changes in expression levels may also be due to different number of copies of DNA constructs that were incorporated in the extra-chromosomal arrays.

Response:

We obtained at least ten independent transgenic lines of *kunEx145[RRP-8(G301R)::GFP]*, and found that all these lines presented significantly decreased expression levels of RRP-8(G301R)::GFP compared with RRP-8::GFP. So we concluded that G301R mutation of RRP-8 showed an obvious decrease in expression.

3. It remains confusing if the 200x or 400x images in Figure 7A were used to yield the quantitative data in Figure 7B. Since the authors conceded that the 200x images were complicated by autofluorescence, they should be removed.

Response: We removed the 200X images from figure 7A in the revised manuscript.

Reviewer #3 (Remarks to the Author):

The authors have made a significant effort to address reviewer's comments, clarify data presentation and add new data that clearly supports their main findings. I believe that the manuscript is now fit for publication.

Once again, we very much appreciate your previous comments and questions, as well

as your strong support on our work. These comments and suggestions were very helpful for improvement of our manuscript.

References for Responses.

- 1 Fumagalli, S. *et al.* Absence of nucleolar disruption after impairment of 40S ribosome biogenesis reveals an rpL11-translation-dependent mechanism of p53 induction. *Nature cell biology* **11**, 501-508, doi:10.1038/ncb1858 (2009).
- 2 Bursac, S. *et al.* Mutual protection of ribosomal proteins L5 and L11 from degradation is essential for p53 activation upon ribosomal biogenesis stress. *Proceedings of the National Academy of Sciences of the United States of America* **109**, 20467-20472, doi:10.1073/pnas.1218535109 (2012).
- 3 Zhang, Y. & Lu, H. Signaling to p53: ribosomal proteins find their way. *Cancer cell* **16**, 369-377, doi:10.1016/j.ccr.2009.09.024 (2009).
- 4 Donati, G. *et al.* Selective inhibition of rRNA transcription downregulates E2F-1: a new p53-independent mechanism linking cell growth to cell proliferation. *Journal of cell science* **124**, 3017-3028, doi:10.1242/jcs.086074 (2011).
- 5 Bensaude, O. Inhibiting eukaryotic transcription: Which compound to choose? How to evaluate its activity? *Transcription* **2**, 103-108, doi:10.4161/trns.2.3.16172 (2011).
- 6 Taylor, R. C. & Dillin, A. XBP-1 is a cell-nonautonomous regulator of stress resistance and longevity. *Cell* **153**, 1435-1447, doi:10.1016/j.cell.2013.05.042 (2013).
- 7 Berendzen, K. M. *et al.* Neuroendocrine Coordination of Mitochondrial Stress Signaling and Proteostasis. *Cell* **166**, 1553-1563.e1510, doi:10.1016/j.cell.2016.08.042 (2016).
- 8 Durieux, J., Wolff, S. & Dillin, A. The cell-non-autonomous nature of electron transport chain-mediated longevity. *Cell* **144**, 79-91, doi:10.1016/j.cell.2010.12.016 (2011).

Reviewer #1 (Remarks to the Author):

The authors have addressed my concerns adequately.

REVIEWERS' COMMENTS:

Reviewer #1 (Remarks to the Author):

The authors have addressed my concerns adequately.

Once again, we very much appreciate all three reviewer's great comments and suggestions, which are very helpful for the improvement of our manuscript.